# Optimistic Critic Reconstruction and Constrained Fine-Tuning for General Offline-to-Online RL

**Qin-Wen Luo**[*1], **Ming-Kun Xie**[*1,2], **Ye-Wen Wang**[*1], **Sheng-Jun Huang**[†1]

[1] Nanjing University of Aeronautics and Astronautics, Nanjing, China
[2] RIKEN Center for Advanced Intelligence Project, Tokyo, Japan
{luoqw8,linuswangg,huangsj}@nuaa.edu.cn
ming-kun.xie@riken.jp

## Abstract

Offline-to-online (O2O) reinforcement learning (RL) provides an effective means of leveraging an offline pre-trained policy as initialization to improve performance rapidly with limited online interactions. Recent studies often design fine-tuning strategies for a specific offline RL method and cannot perform general O2O learning from any offline method. To deal with this problem, we disclose that there are evaluation and improvement mismatches between the offline dataset and the online environment, which hinders the direct application of pre-trained policies to online fine-tuning. In this paper, we propose to handle these two mismatches simultaneously, which aims to achieve general O2O learning from any offline method to any online method. Before online fine-tuning, we re-evaluate the pessimistic critic trained on the offline dataset in an optimistic way and then calibrate the misaligned critic with the reliable offline actor to avoid erroneous update. After obtaining an optimistic and and aligned critic, we perform constrained fine-tuning to combat distribution shift during online learning. We show empirically that the proposed method can achieve stable and efficient performance improvement on multiple simulated tasks when compared to the state-of-the-art methods. The implementation is available at https://github.com/QinwenLuo/OCR-CFT.

## 1 Introduction

Offline reinforcement learning (RL) aims to learn a policy from a fixed dataset without additional interactions with the environment. This characteristic makes it particularly promising for critical applications such as healthcare decision-making [39], human-AI coordination [14] and autonomous driving [8]. Generally, the performance of the learned policy relies on the quality of the dataset. Given that the offline data is limited, fine-tuning the policy through interactions with the environment is still necessary to achieve favorable performance. Consequently, offline-to-online (O2O) RL tends to achieve faster performance improvements based on better initializations.

To effectively fine-tune offline policies, O2O methods are typically designed based on specific offline RL algorithms. Existing methods can be roughly divided into two groups according to the base offline methods they use. The first group relies on policy constraint methods. These approaches aim to improve online performance by either adaptively adjusting constraints [5, 52, 43] or directly applying offline algorithms to online fine-tuning [32, 20]. Unfortunately, these methods often suffer from inefficient performance improvement due to restricted action exploration caused by policy constraints. The second group builds on value regularization methods. These methods aim to prevent excessively

---

[*]The authors contribute equally.
[†]Correspondence to: Sheng-Jun Huang (huangsj@nuaa.edu.cn).

low Q-values resulting from pessimistic evaluations, such as those in CQL [21]. The goal is to enhance the generalization of the value function and mitigate potential performance declines [27, 33]. Some other methods adopt ensemble Q-learning [51, 23, 30, 19] address these issues. However, these methods often face high computational costs due to the need to train multiple Q-networks.

Considering that the aforementioned methods develop online fine-tuning algorithms based on specific offline methods, they often struggle to be applied to other offline methods. To establish a general O2O framework, it is essential to address the core issues associated with transitioning from offline to online environments. Inspired of the recent work [48], which highlights the misalignment between the actor and the critic in an explicit policy constraint method, we identify two mismatches in general O2O RL: evaluation mismatches and improvement mismatches. Evaluation mismatches primarily occur in value regularization methods. These refer to the differences in policy evaluation methods between online and offline settings, which cause severe fluctuations in Q-value estimation during the initial stages of fine-tuning. Improvement mismatches, on the other hand, are prevalent in policy constraint methods. They are often caused by differences in objectives for updating the policy, leading to a misalignment between the probabilities of actions and their Q-values. Thanks to another recent work by Xu et al [47], which connects value regularization and policy constraint methods, we bridge these two types of mismatches within a unified framework for general RL-based offline algorithms.

In this paper, we propose a general O2O framework designed to address both evaluation and improvement mismatches simultaneously, aiming for stable and favorable fine-tuning performance from any offline method to representative online methods. To address the evaluation mismatch in value regularization methods, we propose re-evaluating the offline policy in an optimistic manner using an off-policy evaluation method. This approach allows us to obtain optimistic Q-value estimates, preventing the dramatic fluctuations in Q-values that could potentially cause the policy to collapse. Although the re-evaluated critic can estimate Q-values optimistically, it suffers from the misalignment with the offline policy, causing the improvement mismatch. To handle the improvement mismatch in the re-evaluated critics and policy constraint methods, we introduce value alignment to calibrate the critic so that it aligns with the probabilities predicted by the policy. Our approach involves using the Q-value of the most likely action as an anchor and then calibrating other Q-values by either exploiting the correlation between Q-values of different state-action pairs or modeling Q-values as a Gaussian distribution. Finally, we propose a constrained fine-tuning framework to guide the policy update by adding a regularization term, with the target of mitigating the negative impact of data shift. Extensive experimental results on multiple benchmark environments validate that the proposed methods can achieve better or comparable performance when compared to state-of-the-art methods.

Our contributions can be summarized as follows:

- We systemically study that there exist evaluation and improvement mismatches for offline RL methods from the perspective of online RL. We show that resolving these two types of mismatches is essential for achieving general O2O RL.
- We develop two techniques to address these mismatches. Policy re-evaluation aims to achieve optimistic Q-value estimates, preventing instability in Q-value estimation. Value alignment calibrates the critic to align with the policy, ensuring consistency between action probabilities and their corresponding Q-values.
- We introduce a constrained fine-tuning framework that incorporates a regularization term into the policy objective, combating the inevitable distribution shift and ensuring stable and optimal performance when fine-tuning the policy in online environments.

## 2   Preliminaries

In this section, we introduce necessary preliminaries about RL, including Markov decision process, and three target online RL methods.

**Markov Decision Process (MDP)**   A Markov Decision Process $\mathcal{M}$ is defined by the tuple $(S,A,R,P,\mu,\gamma)$ [38], where $S$ is the state space, $A$ is the action space, $P : S \times A \to \Delta(S)$ is the transition function, $R : S \times A \to \mathbb{R}$ is the reward function, $\mu$ is the initial state distribution, and $\gamma$ is a discount factor. The goal is to learn a policy that maximizes the expected return as

$$J(\pi) = \mathbb{E}_\pi[\sum\nolimits_{t=0}^{\infty} \gamma^t r_t] = \mathbb{E}_{\delta(s_0,a_0)}[Q^\pi(s_0,a_0)], \delta(s_0,a_0) := (s_0 \sim \mu, a_0 \sim \pi(\cdot|s_0)). \quad (1)$$

**Online RL** The three most commonly used online RL algorithms are SAC [17], TD3 [11], and PPO [37]. Below, we will introduce these three methods one by one.

**SAC** first introduces entropy maximization into the RL scenarios and updates the actor and the critic by minimizing the following objectives:

$$\mathcal{L}_\pi^{\mathrm{SAC}}(\theta) = \mathop{\mathbb{E}}_{s \sim R} \mathop{\mathbb{E}}_{a \sim \pi_\theta(\cdot|s)} \left[ \alpha \log \pi_\theta(a|s) - Q_\mu(s,a) \right], \tag{2}$$

$$\mathcal{L}_Q^{\mathrm{SAC}}(\mu_i) = \mathop{\mathbb{E}}_{(s,a,r,s') \sim R} \left[ (Q_{\mu_i}(s,a) - y(r,s'))^2 \right], \tag{3}$$

$$y(r,s') = r + \gamma \mathbb{E}_{a' \sim \pi_\theta(\cdot|s')} \left[ \min_{i=1,2} Q_{\bar{\mu}_i}(s,a) - \alpha \log \pi_\theta(a'|s') \right],$$

$$\mathcal{L}_\alpha^{\mathrm{SAC}}(\alpha) = -\alpha \mathop{\mathbb{E}}_{s \sim R} \mathop{\mathbb{E}}_{a \sim \pi_\theta(\cdot|s)} \left[ \log \pi_\theta(a|s) - \bar{\mathcal{H}} \right], \tag{4}$$

where $\alpha > 0$, $\bar{\mu}_i$ are the parameters of the target $Q$ network, and $\bar{\mathcal{H}}$ is the target entropy.

**TD3** models a deterministic policy and uses tricks, including clipped double Q-learning and policy smoothing, to address the issue of function approximation error. The deterministic policy gradient used to update policy is defined as

$$\nabla_\theta J_\pi^{\mathrm{TD3}}(\theta) = \mathop{\mathbb{E}}_{s \sim R} \left[ \nabla_a Q_\mu^\pi(s,a) |_{a=\pi(s)} \nabla_\theta \pi_\theta(s) \right]. \tag{5}$$

The objective function of updating the critic is defined as

$$\mathcal{L}_Q^{\mathrm{TD3}}(\mu_i) = \mathop{\mathbb{E}}_{(s,a,r,s') \sim R} \left[ (Q_{\mu_i}(s,a) - y(r,s'))^2 \right]. \tag{6}$$

Here, $y(r,s') = r + \gamma \min_{i=1,2} Q_{\bar{\mu}_i}(s', \tilde{a})$, where $\tilde{a} = \pi_{\bar{\theta}}(s') + \epsilon$ and $\epsilon \sim \mathrm{clip}(N(0,\sigma), -c, c)$.

**PPO** provides a simple implementation for TRPO [35], which updates the actor and the critic by minimizing the following objective functions, respectively,

$$\mathcal{L}_\pi^{\mathrm{PPO}}(\theta) = - \mathop{\mathbb{E}}_{s \sim \nu^{\pi_{\theta_k}}} \mathop{\mathbb{E}}_{a \sim \pi_{\theta_k}(\cdot|s)} \left[ \min \left( r(\theta) A^{\pi_{\theta_k}}(s,a), \mathrm{clip}(r(\theta), 1-\epsilon, 1+\epsilon) A^{\pi_{\theta_k}}(s,a) \right) \right], \tag{7}$$

$$\mathcal{L}_V^{\mathrm{PPO}}(\nu) = \mathop{\mathbb{E}}_{(s,r,s') \sim R} \left[ (A^{\pi_{\theta_k}}(s,a) + V_{\bar{\nu}}(s')) - V_\nu(s))^2 \right], \tag{8}$$

where $r(\theta) = \pi_\theta / \pi_{\theta_k}$ and the advantage $A^{\pi_{\theta_k}}$ is computed by GAE [36].

It is noteworthy that previous works mostly adopted SAC and TD3 for online fine-tuning as they are off-policy methods. In our work, given the widespread use of PPO in online RL, we also employ it for fine-tuning though it is an on-policy method.

## 3 Evaluation and Improvement Mismatches

In this section, we focus on the differences between online and offline RL and study how these differences impact the performance of online fine-tuning. In general, we summarize these differences as two types of mismatches, *evaluation mismatch* and *improvement mismatch*. The former arises from the changes in policy evaluation functions during the transition from offline to online environments; while the latter represents the inconsistency in the objectives for policy updates between offline and online RL. Most offline RL methods suffer from one or both of these issues, which underscores the importance of addressing these mismatches to achieve stable and effective online fine-tuning.

Evaluation mismatch often occurs in the value regularization methods. For example, in CQL [21], a representative offline method, the policy evaluation function transitions from a pessimistic estimation inherent to offline learning to a more optimistic estimation during online training. This shift frequently results in a sharp increase in Q-values at the beginning of online fine-tuning, which can hinder stable performance improvements. To address this problem, several attempts have been made to mitigate the excessive underestimation of out-of-the-distribution (OOD) actions during offline learning [33, 27]

or to initialize a pessimistic Q-ensemble to maintain pessimism during online fine-tuning [23]. Unfortunately, these approaches are predominantly tailored to the transition from CQL to SAC, limiting their applicability to other offline algorithms.

Improvement mismatch is commonly found in policy constraint methods. Typical examples include TD3+BC [10] and AWAC [32], where the objective of actor updates differs significantly from typical online methods. In these models, updates to the actor are not solely reliant on the critic's evaluation. Consequently, actions that have high Q-values may not automatically translate to high probabilities of being selected, and vice versa. This divergence often misguides the update of policy at the beginning stage of online fine-tuning, resulting in unfavourable performance.

Besides, One-step RL and non-iterative methods, *e.g.*, IQL, exhibit both evaluation and improvement mismatches. During the policy evaluation stage, these methods estimate the Q-values based on the behavior policy or an unknown policy [18] instead of the target policy as in online methods; during the improvement stage, because they impose additional constraints on policy updates, they also encounter the same issue as discussed above. This can lead to discrepancies in both the assessment of action values and the subsequent policy optimization, making it hard to achieve effective policy improvement.

Thanks to the recent work [47], which presents a unified framework for understanding offline RL, we bridge these two mismatches for general RL-based offline algorithms. Formally, the offline RL problem can be defined by the *behavior-regularized* MDP problem [47, 13] via maximizing the following objective:

$$\mathbb{E}_\pi \left[ \sum_{t=0}^\infty \gamma^t \left( r(s_t, a_t) - \alpha \cdot f \left( \frac{\pi(a_t|s_t)}{\mu(a_t|s_t)} \right) \right) \right] = \mathbb{E}_{\delta(s_0, a_0)} \left[ Q^\pi(s_0, a_0) - \alpha \cdot f \left( \frac{\pi(a_0|s_0)}{\mu(a_0|s_0)} \right) \right],$$
(9)

where $f(\cdot)$ is a regularization function and $\mu$ is the behavior policy.

From Eq. (1) and Eq. (9), we observe a significant divergence in the relation between the actor and the critic in online versus offline scenarios. Unlike in online cases where the policy update is solely dependent on the Q-function, in offline scenarios, it also critically depends on the data distribution, as indicated by the regularization term in Eq. (9). This distinction highlights why offline methods often face evaluation and improvement mismatches when applied to online fine-tuning. Using offline actor and critic trained by Eq. (9) for initialization in online fine-tuning introduces both types of mismatches, resulting in unstable and inefficient updates.

## 4  Method

In this section, we introduce our proposed O2O method for handling the mismatches discussed in Section 3 and the distribution shift problem. To address the pessimistic or unreliable evaluation, such as in CQL and IQL, we develop a policy re-evaluation technique. This technique optimistically re-evaluates the well-trained offline policy using an off-policy evaluation method. Although this re-evaluation helps the critic achieve more optimistic Q-value estimates, unavoidable factors such as function approximation errors and partial data coverage can still lead to a misalignment between the critic and the offline policy. This misalignment means that the action with the highest probability predicted by the policy does not necessarily have the highest Q-value, leading to what we have termed improvement mismatch.

As discussed in in Section 3, both the value regularization (after re-evaluation) and policy constraint methods exhibit improvement mismatch, though the reasons for the mismatch differ between these two approaches. To address this issue, we propose value alignment, which aims to align the critic's estimates with the policy's action probabilities, effectively tackling the improvement mismatch in both types of methods. Finally, to deal with the inevitable distribution shift between offline and online environments, we develop a constrained fine-tuning framework. This framework ensures that the policy consistently updates in the optimal direction by incorporating a regularization term into the policy objective.

We propose methods for various representative online RL algorithms within a unified framework, including SAC [17], TD3 [11], and PPO [37]. These algorithms represent the mainstream approaches in online RL, and are targeted respectively for the off-policy approach with stochastic policies, the off-policy approach with deterministic policies, and the on-policy approach.

## 4.1 Policy Re-evaluation

The critic trained on an offline dataset typically maintains pessimistic estimates of Q-values. When using online evaluation methods to fine-tune this critic without any value regularization, the Q-values can experience a dramatic jump, especially for OOD actions, leading to inaccurate Q-value estimations. To mitigate this problem, we propose re-evaluating the offline policy to acquire a new critic by employing an off-policy evaluation (OPE) method. The goal is to enable the critic to have optimistic estimates of Q-values that more closely approximate the true values.

However, directly applying OPE methods for policy evaluation on offline datasets often leads to large extrapolation errors, as discussed in the previous work [12]. These errors arise due to absent data and training mismatches. Thanks to the pessimism in offline RL, it is reasonable to assume that a well-trained policy is close enough to the behavior policy or even captures the support set of the behavior policy. A common assumption is single-policy concentrability [45, 34], which demonstrates how concentrated a learned policy is within the given dataset and can be defined as follows.

**Assumption 4.1.** *($\pi_\theta$-concentrability [45]) The behavior policy $\mu$ and learned policy $\pi_\theta$ satisfy*

$$\max_{(s,a) \in S \times A} \frac{d^{\pi_\theta}(s,a)}{d^\mu(s,a)} \leq C.$$

where $d^{\pi_\theta}(s,a)$ is the occupancy measure of $\pi_\theta$ and $C$ is a constant. Single-policy concentrability measures the degree to which the state-action distribution induced by the learned policy is covered by the dataset used for training. By ensuring that the learned policy does not deviate significantly from the behavior policy, the extrapolation error can be greatly reduced [22, 29]. When using a representative OPE method called fitted Q-evaluation (FQE), based on Assumption 4.1 and Theorem 4.2 in [22], the upper bound of extrapolation error can be obtained.

**Corollary 4.2.** *Under Assumption 4.1, by denoting Q-value function class as $\mathcal{F}$, for $\delta \in (0,1)$, after $K$ iterations of FQE on the dataset $\mathcal{D}$, with probability $1 - \delta$, we have:*

$$|Q^\pi - \hat{Q}^\pi| \leq \frac{1-\gamma^K}{1-\gamma}\sqrt{C\epsilon} + \gamma^K \bar{V}, \text{ where } \epsilon := \frac{22\bar{V}^2 \log(|\mathcal{F}|/\delta)}{|\mathcal{D}|} + 20d_F^\pi.$$

where $d_F^\pi$ is inherent Bellman evaluation error (Definition 4.1 in [22]), and $\bar{V}$ is the maximum of $V$, which can be bounded by $R_{max}/(1-\gamma)$.

With a powerful neural network and sufficient data, the inherent Bellman evaluation error could be tiny. Accordingly, with a large training step $K$, the error will be bounded by an acceptable value. This implies that, given sufficient data, one can achieve a critic with optimistic property and minor extrapolation error through policy re-evaluation. In practical implementation, for off-policy methods SAC and TD3, we can directly use Eq. (3) and Eq. (6) to re-evaluate the offline policy. For the on-policy method PPO, we train a critic by fitting the returns of offline trajectories. Considering that the critic can only approximate $V^\mu(s)$ rather than the true value function, we propose a regularization term in Section 4.2, which ensures the critic's estimates are reliable and conducive to effective policy improvement.

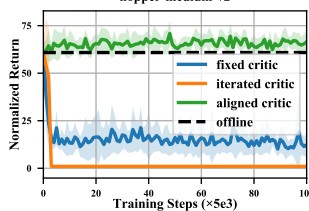

(a) The performance of SAC

## 4.2 Value Alignment

Although the critic after policy re-evaluation possesses the optimistic property needed in the online environment, it often does not align well with the offline policy due to factors such as function approximation errors, generalization errors of neural networks, and partial data coverage. Moreover, as discussed in Section 3, policy constraint offline methods also suffer from misalignment between the critic and policy. The misalignment means that the

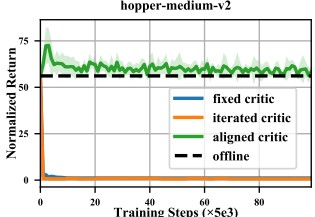

(b) The performance of TD3

Figure 1: The results of actors updated with different critics.

action with the highest Q-value does not necessarily have the highest probability, often leading to misleading updates of the policy. To verify this observation, we trained the policy using SAC and TD3

with three different critics: the fixed re-evaluated critic, the iterative re-evaluated critic (which updates with the policy), and our aligned critic. From Figure 1, the performance of re-evaluated critics sharply declines at the initial stage and does not recover in the subsequent training; while our aligned critic achieves stable and favorable performance. These results indicates that the misalignment between the re-evaluated critic and the offline policy can make it difficult for the policy to optimize in a correct direction, leading to an irreversible decline in performance. Given that the well-trained offline policy is reliable, the desirable critic should not only have optimistic Q-value estimates but also maintain alignment with the offline policy. To achieve this, we propose performing value alignment to calibrate the critic. The main idea is to use the Q-values of the offline policy actions as anchors, keeping them unaltered, and to suppress any overestimated Q-value that exceed these anchors, thereby aligning the critic with the offline policy. Below, we will discuss how to implement value alignment for different online methods.

**O2SAC**   Recall that in SAC [16, 17], the optimal policy is defined as

$$\pi(a|s) = \exp(\frac{1}{\alpha}Q(s,a))/\exp(\frac{1}{\alpha}V(s)) \tag{10}$$

With a simple transformation of Eq. (10), we have

$$Q(s,a) = V(s) + \alpha \log \pi(a|s) \tag{11}$$

One intuition behind our method is that actions with higher probabilities typically have more accurate Q-value estimates because these actions are closer to the dataset, and there is sufficient data nearby to obtain a precise estimate. This motivates us to use the Q-value $Q_{\bar{\mu}}(s, \dot{a})$ of the optimal action $\dot{a}$ to calibrate any other overestimated action. Assuming that $\pi_{\text{off}}$ is the offline policy, we perform value alignment for any state-action pair $(s, a)$ as follows:

$$Q'_{\mu}(s,a) = \min\left(Q_{\bar{\mu}}(s,\dot{a}) - \alpha\left(\log \pi_{\text{off}}(\dot{a}|s) - \log \pi_{\text{off}}(a|s)\right), Q_{\bar{\mu}}(s,a)\right), \tag{12}$$

where $\min(\cdot, \cdot)$ operator is used to maintain the Q-values of actions that are not overestimated and consistent with OPE results.

Formally, we define the objective function of value alignment as follows:

$$\begin{aligned}
\mathcal{L}_Q^{\text{align}}(\mu_i) &= \mathop{\mathbb{E}}_{s\sim R, a\sim \pi(\cdot|s)}\left[\left(Q_{\mu_i}(s,a) - Q'_{\mu}(s,a)\right)^2\right] \\
\mathcal{L}_Q^{\text{retain}}(\mu_i) &= \mathop{\mathbb{E}}_{s\sim R,}\left[\left(Q_{\mu_i}(s,\dot{a}) - Q_{\bar{\mu}}(s,\dot{a})\right)^2|_{\dot{a}=\pi_{\text{off}}(s)}\right] \\
\mathcal{L}_Q^{\text{critic}}(\mu_i) &= \mathcal{L}_Q^{\text{align}}(\mu_i) + \mathcal{L}_Q^{\text{retain}}(\mu_i)
\end{aligned} \tag{13}$$

Considering that the overall Q-values have decreased, to maintain an optimistic estimate, we use the regularization term $\mathcal{L}_Q^{\text{retain}}$ to prevent the underestimation of Q-values.

We derive Proposition 4.3 to show that the aligned state values $V_{\text{align}}(s)$ can be maintained within an appropriate range, meaning that the estimates remain optimistic while avoiding overestimation.

**Proposition 4.3.** *After the value alignment process of Eq. (13), the state value function $V_{align}(s)$ satisfies*

$$V_{fqe}(s) \leq V_{align}(s) \leq V_{\dot{a}}(s) \tag{14}$$

*where $Q_{fqe}(s, a)$ are the low Q-values after policy re-evaluation, which do not require calibration, $V_{fqe}(s) = Q_{fqe}(s,a) - \alpha \log \pi(a|s)$ and $V_{\dot{a}}(s) = Q(s,\dot{a}) - \alpha \log \pi(\dot{a}|s)$.*

During value alignment, we update the policy with Eq. (2) simultaneously for sampling the overestimated actions. The whole process iterates Eq. (2) and Eq. (13) to obtain a policy that performs equivalently to the offline policy and a critic aligned with it. We denote the policy as $\pi_{\text{on}}$ and the aligned critic as $Q_{\text{on}}$ for sequential training. Note that $Q_{\text{on}}$, which is modified from the critic obtained in the policy re-evaluation, does not depend on specific offline critics. This flexibility allows us to implement the transition to SAC from different offline algorithms.

**O2TD3**   As done in the case of O2SAC, we can use the Q-values of the policy actions $\dot{a}$ to maintain the optimistic property of policy re-evaluation and calibrate the Q-values of other actions.

Unfortunately, in TD3, the actor is modeled as a deterministic policy, lacking an explicit expression for Q-values, which prevents us from directly aligning Q-values with the policy.

To solve this problem, our main idea is to model the distribution of Q-values around the policy action $\dot{a}$ as a Gaussian distribution. Specifically, in Eq. (5), when we use the offline policy as $\pi_\theta$, the gradient of the policy is only related to the the gradient of $Q(s, a)$ with respect to $a$. It is easy to see that the gradient around $\dot{a}$ should tend to 0 as $\dot{a}$ is the optimal action selected by the offline policy. Due to the use of smoothing regularization in the critic's update in TD3, the Q-values of actions near $\dot{a}$ differ only slightly from the Q-value of $\dot{a}$ itself. This enables us to assume that normalized Q-values around $\dot{a}$ follow a Gaussian distribution $Q(s, a)/Q(s, \dot{a}) \sim N(\dot{a}, \Sigma)$. Formally, we can calibrate the Q-values of other actions as (see Appendix G.2 for detailed derivation)

$$Q'(s, a) = \min \left( Q_{\bar{\mu}}(s, \tilde{a}), \frac{Q(s, \dot{a})}{1 + k \cdot \max\left(d(a, \dot{a})^2, \sigma^2\right)} \right) \tag{15}$$

where $k$ is a constant, which is set as 1 across all tasks, and $d(a, \dot{a})$ is a distance measure, which is defined as the euclidean distance divided by the square root of the action dimension in our implementation.

From Eq. (15), after calibration, the Q-values of the actions around $\dot{a}$ are only slightly lower than $Q(s, \dot{a})$, which ensures that $\dot{a}$ is the output action of the policy while maintaining a smoothing and optimistic property of Q-values. Moreover, for the actions that differ greatly from $\dot{a}$, we limit the maximum of the distance measure $d(a, \dot{a})$ in Eq. (15) to the policy noise used in Eq. (6) to avoid severe underestimation of their Q-values. Formally, we define the objective loss of value alignment for O2TD3 as

$$\mathcal{L}_Q^{\text{align}}(\mu_i) = \mathbb{E}_{s \sim R} \left[ \left( Q_{\mu_i}(s, \tilde{a}) - Q'_\mu(s, \tilde{a}) \right)^2 \big|_{\tilde{a} = \pi(s) + \delta} \right]$$

$$\mathcal{L}_Q^{\text{retain}}(\mu_i) = \mathbb{E}_{s \sim R} \left[ \left( Q_{\mu_i}(s, \dot{a}) - Q_{\bar{\mu}}(s, \dot{a}) \right)^2 \big|_{\dot{a} = \pi_{\text{off}}(s)} \right] \tag{16}$$

$$\mathcal{L}_Q^{\text{critic}}(\mu_i) = \mathcal{L}_Q^{\text{align}}(\mu_i) + \mathcal{L}_Q^{\text{retain}}(\mu_i)$$

where $Q'_\mu(s, \tilde{a}) = \min(Q_{\bar{\mu}}(s, \tilde{a}), Q'_\mu(s, \tilde{a}))$ and $\tilde{a}$ is a perturbed action defined in Eq (6).

**O2PPO**   In PPO, only the critic $V(s)$ is used to estimate the advantages for policy update. During the re-evaluation process, we train a critic by fitting the returns of offline trajectories as mentioned in Section 4.1. This indicates that the re-evaluated critic only approximate $V^\mu(s)$ instead of the true one, misguiding the update of the policy. To mitigate this problem, we propose an auxiliary advantage function to correct erroneous updates.

Generally, a desirable auxiliary advantage function should satisfy the following two conditions: 1) it enables the policy to update in a reliable region; 2) its value must be zero at the beginning of online fine-tuning to enable the policy to transition smoothly from offline to online. Considering that the well-trained offline policy is reliable, we define the auxiliary advantage function as

$$A_\alpha(s, a) = \alpha \log \pi_{\text{off}}(a|s) + \alpha \mathcal{H}(\pi_{\text{off}}(\cdot|s)) \tag{17}$$

where $\mathcal{H}$ is the entropy of action probabilities predicted by $\pi_{\text{off}}$. It is easy to verify the second condition that $A_\alpha(s, a) = 0$ at the beginning of offline fine-tuning. To verify the first condition, we derive the following proposition. Its proof can be found in Appendix F.

**Proposition 4.4.** *With $A_\alpha(s, a)$ in Eq.* (17)*, the policy update is regularized by the cross-entropy loss about the offline policy, thereby constraining the policy update in a reliable region.*

Accordingly, we define the advantage function for policy update as

$$A'(s, a) = A(s, a) + \beta A_\alpha(s, a) \tag{18}$$

where $\beta$ anneals to 0 from 1. With the auxiliary advantage, Eq. (18) prevent the update direction from deviating too far from the offline policy.

## 4.3   Constrained Fine-Tuning

In the previous subsections, we discussed how to address the mismatch issues in the O2O problems. However, due to the distribution shift between the offline dataset and the online environment, along

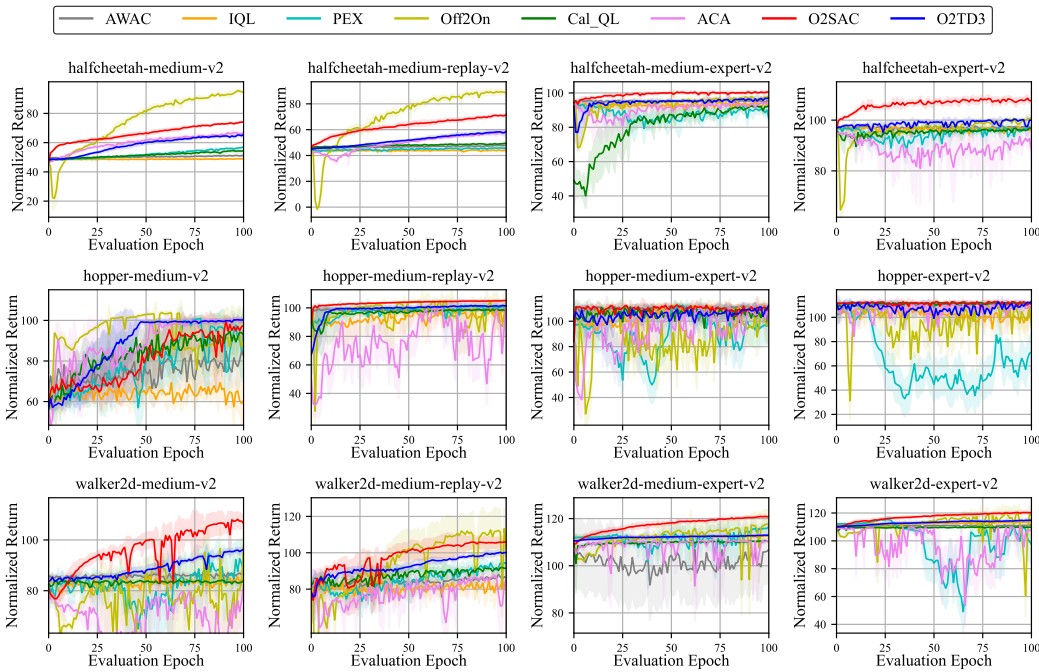

Figure 2: Performance curves on D4RL [9] MuJoCo locomotion tasks during online fine-tuning.

with the optimism in online RL, encountering out-of-distribution (OOD) states and actions becomes inevitable, potentially leading to significant performance fluctuations. Especially for OOD states that are absent in the offline dataset, even though the policy was trained well in the offline phase, it may still fail to output favourable actions, thereby causing erroneous policy update.

Considering that the critic maintains an optimistic nature after undergoing policy re-evaluation and value alignment, with the optimistic update way during online fine-tuning, it typically overestimates the Q-values of OOD state-action pairs, which can mislead the update of the policy. Inspired of CMDP [1, 41, 7], we develop constrained fine-tuning (CFT) to introduce a regularization term to constrain the current actor and critic, which prevents the policy update from being severely misguided.

Specifically, we impose a constraint term $f(\pi, \pi_{\text{ref}})$ on the policy objective to ensure that it updates within the credible region of $\pi_{\text{ref}}$, where $f(\cdot, \cdot)$ is a divergence measurement and $\pi_{\text{ref}}$ is the optimal historical policy during online evaluations. Formally, we define the policy objective of CFT as

$$\max \mathbb{E}_\pi\Big[\sum_{t=0}^{\infty} \gamma_t r_t(s_t, a_t)\Big] \quad \text{s.t. } \mathbb{E}_\pi[f(\pi(a_t|s_t), \pi_{\text{ref}}(a_t|s_t))] < \tau \tag{19}$$

By incorporating the constraint term into the reward function akin to RCPO [41], we can solve the problem by minimizing the following objective functions with initializing $\pi_{\text{ref}}$ as $\pi_{\text{on}}$ obtained in Section 4.2 at the beginning of online fine-tuning.

$$
\begin{aligned}
L_\pi(\theta) &= \max \mathbb{E}_{\pi_\theta}[Q_\mu^{\pi_\theta}(s, a) - \lambda f(\pi_\theta(a|s), \pi_{\text{ref}}(a|s))] \\
L_Q(\mu) &= \min \mathbb{E}_{(s,a,r,s')\sim R}[(Q_\mu^{\pi_\theta}(s, a) - y)^2] \\
y &= r + \gamma \mathbb{E}_{a'\sim\pi_\theta(\cdot|s')}[Q_{\bar{\mu}}^{\pi_\theta}(s', a') - \lambda f(\pi_\theta(a'|s'), \pi_{\text{ref}}(a'|s'))] \\
L(\lambda) &= \min_{\lambda\geq 0} -\lambda\left[\mathbb{E}_{\pi_\theta}(f(\pi_\theta(a|s), \pi_{\text{ref}}(a|s))) - \tau\right]
\end{aligned}
\tag{20}
$$

We provide a theoretical guarantee for the proposed CFT framework.

**Corollary 4.5.** *With the penalty $f(\pi, \pi_{ref})$ defined before and appropriate learning rates, algorithm of Eq. (20) almost surely to a fixed point $(\theta^\star, \mu^\star, \lambda^\star)$, where $\lambda^\star = 0$, $\theta^\star$ and $\mu^\star$ are corresponded to $\pi^\star$ and $Q^\star$, which are optimal in the MDP without constraint.*

In our implementation, we use KL divergence and MSE function as $f(\cdot, \cdot)$ for the transitions of O2SAC and O2TD3, respectively. For O2PPO, the auxiliary advantage function already has the

Table 1: Average normalized D4RL scores of our methods shown in AntMaze navigation tasks after 200k interactions with the environment. (U=umaze, D=diverse)

| Dataset | IQL | PEX | Cal-QL | O2TD3 | O2SAC | O2PPO |
|---------|-----|-----|--------|-------|-------|-------|
| U-v2 | 80.8→80.8 | 85.0→96.2 | 80.8→97.0 | 92.8→95.8 | 92.8→93.6 | 77.3→98.0 |
| U-D-v2 | 56.6→35.8 | 12.6→16.0 | 23.8→71.2 | 38.4→52.2 | 43.8→79.8 | 56.4→86.3 |
| total | 137.4→116.6 | 97.6→112.2 | 104.6→168.2 | 128.6→142.0 | 131.2→148.0 | 133.7→184.3 |

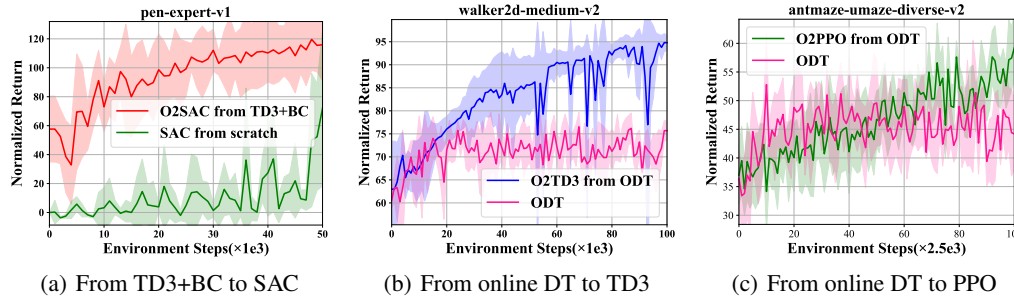

(a) From TD3+BC to SAC      (b) From online DT to TD3      (c) From online DT to PPO

Figure 3: The fine-tuning performance achieved by transferring to three online algorithms from their heterogeneous offline algorithms.

ability to constrain the policy update, we only need to replace $\pi_{\text{off}}$ with $\pi_{\text{ref}}$ during online fine-tuning. Note that in our methods, at the beginning of online fine-tuning, the regularization function $f(\pi_\theta(a|s), \pi_{\text{on}}(a|s))$ is zero for any state, which guarantees no destruction of the alignment for the actor and the critic obtained in Section 4.2.

## 5 Experiments

In this section, we perform experiments to validate the effectiveness of the proposed method on D4RL [9] MuJoCo and AntMaze tasks, including HalfCheetah, Hopper, Walker2d and AntMaze environments. Specifically, we compare our methods with AWAC [32], IQL [20], PEX [50], Off2On [23], Cal-QL [33] and ACA [48]. For all methods except for O2PPO, we run 100,000 interaction steps with the environment and evaluate the policy per 1000 steps, as they all use off-policy methods for online fine-tuning. For our O2PPO method, due to the low efficiency of the on-policy method, we run 250,000 interaction steps to validate its performance, with 2500 steps as the evaluation interval. We run all methods with five random seeds and report their averaging results. Due to the space limitation, more experimental results can be found in Appendix B and C.

### 5.1 Comparison with State-of-the-Art Methods

We respectively initialize the policies of O2SAC, O2TD3 and O2PPO from the results of CQL, TD3+BC and IQL. Figure 2 shows the performance curves of our methods and comparing methods on Mujoco tasks. Similar to the previous works [48, 5], for O2SAC and O2TD3, we use CQL and TD3+BC for offline policy pre-training. From the figure, we can see that our method can converge more stably and rapidly than other methods and achieve the optimal performance in most cases. Although the ensemble method Off2On can achieve better final performance than our methods in some cases, it often suffers from a dramatic drop in performance at the beginning of fine-tuning, which is often unacceptable in the O2O problems. We report the results of O2PPO in Appendix A because of its different number of interaction steps. Although PPO is an on-policy method with low efficiency, from the results in Table 1, it shows significant superiority on sparse reward tasks, even though with equal interactions.

### 5.2 Study on Transferability

In this section, we perform experiments to verify the powerful transferability of the proposed method. As mentioned earlier, one of the advantages of our method is that it imposes no requirements on offline

algorithms. This means we can achieve the transfer from any offline RL algorithm to three online RL algorithms. Figure 3 illustrates the fine-tuning performance of three online algorithms transferring from their heterogeneous offline algorithms. From Figure 3(a), pre-trained with TD3+BC, O2SAC outperforms SAC trained from scratch with a larger margin. Although Online Decision Transformer (ODT) achieves favourable performance in the offline environment, it converges slowly during online fine-tuning due to the architecture of the transformer. Our O2TD3 and O2PPO significantly enhances its performance through online fine-tuning. These results convincingly verify that our method show the strong transferability from various offline methods.

## 6  Conclusion

In this paper, we disclose there exist two types of mismatches when online fine-tuning offline RL methods. To address these two mismatches in O2O RL, we proposed optimistic critic reconstruction to re-evaluate an optimistic critic and align it with the offline actor before online fine-tuning, ensuring the stable performance improvement at the beginning stage and potentially better efficiency due to the reconstructed optimism consistent with online RL. Furthermore, to combat the inevitable distribution shift that can hinder the stable performance improvement, we introduce constrained fine-tuning to constrain the divergence of current policy and the best foregoing policy to maintain the stability of online fine-tuning. These two components form a versatile O2O framework, allowing the transition from any offline algorithms to three state-of-the-art online algorithms. Experiments show our framework can converge to optimal performance without affecting the aligned critic at the beginning of online fine-tuning and achieve strong empirical performance.

## Acknowledgements

This work was supported by the NSFC (62222605), the Natural Science Foundation of Jiangsu Province of China (BK20222012, BK20211517), and the National Science and Technology Major Project (2020AAA0107000).

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

## A    Detailed Data

Table 2: Average normalized D4RL scores of O2O methods shown in Figure 2. Outside parenthesis: scores at the end of 100k online steps. Inside parenthesis: the increase of that score upon the end of offline training. (M=medium, R=replay, E=expert)

| Dataset | Score($\delta$) | | | | |
|---------|---------|--------|-----|------|------|
|         | Off2On  | Cal-QL | ACA | O2TD3 | O2SAC |
| Hc-M | 94.22(47.37) | 54.36(6.81) | 66.59(20.09) | 65.58±0.7(17.45) | 74.18±0.5(27.32) |
| Ho-M | 90.34(29.83) | 92.83(33.92) | 99.76(41.08) | 100.25±1.1(43.95) | 97.29±8.2(38.0) |
| Wa-M | 98.2(17.55) | 83.93(1.06 ) | 82.55(5.18) | 96.32±1.6(12.71) | 106.51±2.1(25.42) |
| Hc-M-R | 88.74 (43.18) | 49.36(3.31) | 57.93(15.71) | 58.58±1.8(14.31) | 71.65±0.9(26.37) |
| Ho-M-R | 103.61(4.93) | 98.64(2.58) | 102.19(50.3) | 101.29±1.4(34.22) | 105.12±1.2(16.15) |
| Wa-M-R | 102.43(20.78) | 91.61(9.36) | 85.65(8.86) | 100.08±2.2(21.64) | 106.05±2.8(28.75) |
| Hc-M-E | 98.33(2.67) | 92.67(43.62) | 93.54(0.11) | 97.07±1.1(4.98) | 100.41±0.8(6.24) |
| Ho-M-E | 99.47(14.65) | 108.57(4.57) | 109.72(14.17) | 112.49±0.9(10.8) | 107.4±6.0(23.25) |
| Wa-M-E | 118.01(8.51) | 110.3(1.21) | 110.36(2.92) | 112.97±0.4(2.43) | 120.95±0.6(11.51) |
| Hc-E | 100.99(4.59) | 96.9(1.01) | 87.93(-6.65) | 99.96±0.4(3.15) | 107.74±0.8(11.08) |
| Ho-E | 92.13(-19.44) | 110.84(5.1) | 107.79(-1.98) | 112.4±0.5(0.66) | 112.76±0.4(0.95) |
| Wa-E | 118.02(8.3) | 109.71(0.76) | 108.2(0.21) | 114.67±1.1(4.52) | 120.35±0.8(10.5) |
| Total | 1204.5(182.93) | 1099.73(113.3) | 1112.22(150.0) | 1171.66(170.82) | **1230.41(225.54)** |

| Dataset | Score($\delta$) | | | |
|---------|------|-----|-----|-------|
|         | AWAC | IQL | PEX | O2PPO |
| Hc-M | 51.11(1.43) | 48.85(0.33) | 57.05(8.53) | 59.96±0.8(11.54) |
| Ho-M | 82.42(16.53) | 60.48(-1.62) | 95.7(33.59) | 100.42±1.1(48.61) |
| Wa-M | 87.03(1.50) | 83.87(1.34) | 88.18(5.65) | 86.65±2.2(9.0) |
| Hc-M-R | 47.81(2.03) | 43.42(0.32) | 45.44(2.34) | 46.76±1.1(4.22) |
| Ho-M-R | 98.86(0.07) | 95.56(5.16) | 101.75(11.35) | 96.87±4.1(18.14) |
| Wa-M-R | 86.16(6.49) | 84.75(5.42) | 94.1(14.78) | 90.43±5.1(11.6) |
| Hc-M-E | 95.37(0.57) | 92.54(0.26) | 84.99(-7.3) | 89.61±1.1(-3.29) |
| Ho-M-E | 109.76(2.53) | 108.29(11.23) | 97.17(0.11) | 107.61±5.4(8.98) |
| Wa-M-E | 106.87(1.23) | 112.57(0.67) | 116.47(4.57) | 121.4±0.8(9.35) |
| Hc-E | 97.16(0.42) | 96.24(-0.56) | 96.69(-0.1) | 96.2±0.7(2.07) |
| Ho-E | 109.88(-1.0) | 97.15(-5.33) | 71.37(-31.12) | 113.08±0.4(9.09) |
| Wa-E | 111.31(0.64) | 113.43(1.31) | 97.54(-14.57) | 117.49±0.9(9.27) |
| Total | 1083.74(31.44) | 1037.15(18.53) | 1046.45(27.83) | 1126.48(138.58) |

Compared with the baseline methods, our methods outperform much in D4RL Mujoco locomotion tasks, as showned in Table 2. We can compare the fine-tuning performance in groups based on online update way. Off2On, Cal-QL, ACA and our O2SAC are updated in the SAC way during online fine-tuning. Although there is a lack of the baseline methods that update the policy in the TD3 way, we compare our O2TD3 with a recent O2O method PROTO+TD3 [25] in Appendix C.4 to demonstrate the superiority of our methods. And we compare our O2PPO with the policy constraint methods AWAC, IQL and PEX, that are implemented in the IQL way, since the idea of PPO is similar to a kind of policy constraint.

With such groups for comparison, our methods get the best performance improvements respectively. Note that in our implementation, Off2On achieves much better performance than the original paper [23] and the implementations in other papers [48] and [50]. However, our O2SAC still outperforms it with less computational cost during online fine-tuning and less requirements for offline policy. In

addition, our methods bridge different offline algorithms and three SOTA online algorithms, and permit additional modifications for improved algorithms based on the online algorithms.

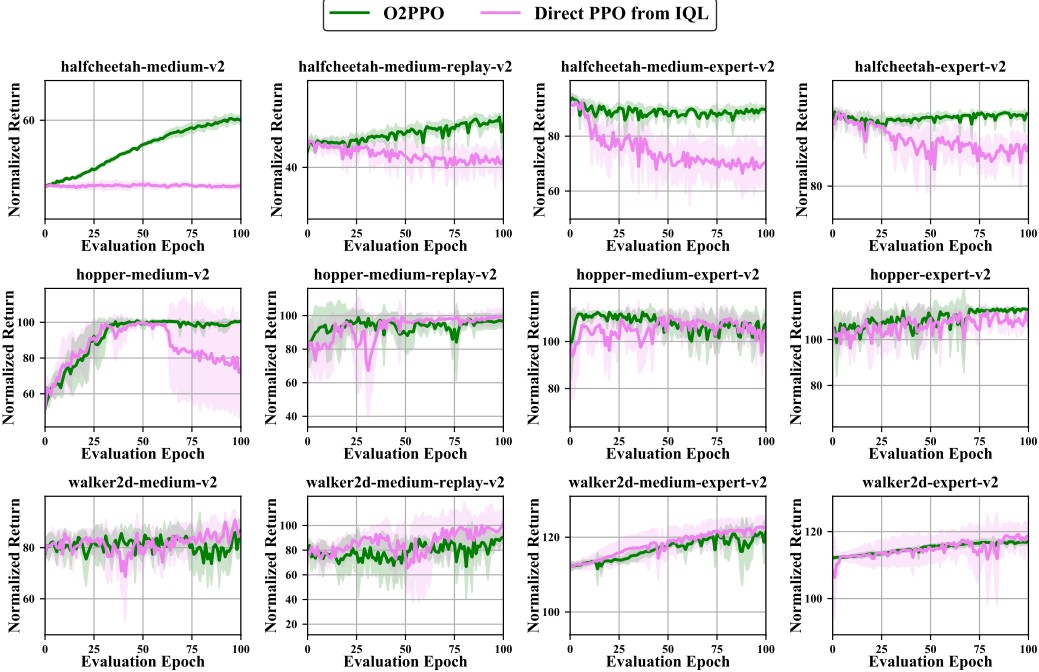

Figure 4: Performance of our O2PPO and direct PPO from IQL on D4RL [9] MuJoCo locomotion tasks during online fine-tuning. The solid lines and shaded regions represent mean and standard deviation.

Due to the low sample efficiency of PPO, we run 250,000 interaction steps for the implementation of O2PPO, which makes it unreasonable to compare the results of different O2O methods in one graph. So we demonstrate the results of O2PPO without any baseline method, but compare our method with the direct way of fine-tuning the offline policy of IQL in the PPO way in the online phase. Note that our purpose is to achieve stable performance improvement, the efficiency depends mainly on the online algorithm, so in some environments, such as *walker2d-medium-replay-v2*, the performance improvement may be less than the direct fine-tuning. This phenomenon could arise due to the critic's capability to approximate the genuine values effectively, thereby facilitating accurate evaluation of the policy, obviating the necessity for corrective adjustments in the policy update direction. And our O2PPO uses an auxiliary advantage function to constrain the policy update, resulting in slower performance improvement. However, since we cannot judge when the critic is reliable, it is necessary to use O2PPO to achieve stable performance improvement, and there is no remarkable discrepancy between O2PPO and the direct way, although in the above scenarios. In general, our O2PPO can achieve stable and efficient performance improvement with limited interactions.

## B  Ablation Study

Before the analysis of ablation studies, we need to clarify the roles and applicability of the different components of our methods for different offline methods.

**The benefit and applicability of policy re-evaluation** As talked before, the purpose of policy re-evaluation is to get optimistic Q-value functions, avoiding the drop of Q-value due to underestimated Q-values in offline. Many previous works [27, 33] discussed such a problem and solve it by alleviating excessive pessimism. Our methods do not need initialize critic from offline results, but re-evaluate the policy in the online way to get optimistic Q-value functions, which is applicable to different algorithms and avoids the performance drop caused by the drastic drop of Q-value. Note that optimistic Q-value functions obtained by policy re-evaluation may be unreasonable, especially for OOD actions, hence value alignment is needed for redress overestimated values.

In some policy constraint methods [32, 10], only the policy is limited to update in a region close to the dataset, but the critic is obtained in the same way as the online update. Therefore, for such offline methods, the online critic can be initialized directly by the offline one, and just need to align the values with the offline policy. In addition, policy re-evaluation can be applied to the scenarios where only the offline policy is provided, such as where the offline policy is obtained in the style of sequence modeling [6, 53, 18] or non-standard RL [46].

**The necessity of value alignment** With the optimistic property, Q-value functions obtained in policy re-evaluation will overestimate OOD actions and induce the policy to take such actions, resulting in performance degradation. For the transition from those policy constraint methods that do not require re-evaluation of the policy, this case still holds. The purpose of value alignment is to approximate the optimistic property and suppress the Q-values of OOD actions to a reasonable estimation. For those offline algorithms where the actor and the critic are misaligned like explicit policy constraint methods, if the constraint is removed during online fine-tuning, the critic will induce actor to align with it. However, the actor is reliable but the critic is not, so such process leads to unknown performance changes that are most likely to be worse. In addition, for algorithms induced by *behavior-regularized* MDP, such problem still exists because the update way between offline and online changes, leading to drastic variation of the critic. Therefore, for O2O RL, it is necessary to align the critic with the actor instead of aligning the actor with the critic.

**The necessity of constrained fine-tuning** Although we keep the optimistic property for Q-value functions and align the critic with actor, it is still challenging to achieve stable online fine-tuning. In general, most of the current offline algorithms focus on how to avoid OOD actions and train a reliable policy on the states of the dataset. Due to the optimism in online RL, OOD states and actions are inevitable, may leading to drastic performance fluctuations, which is undesirable for important scenarios especially for high-risk scenarios. Especially for OOD states, even trained well in the offline phase, the policy still fails to output favourable actions, that may causes erroneous policy update. Therefore, for stable O2O RL, online fine-tuning with the constraint of ensuring safe exploration is necessary.

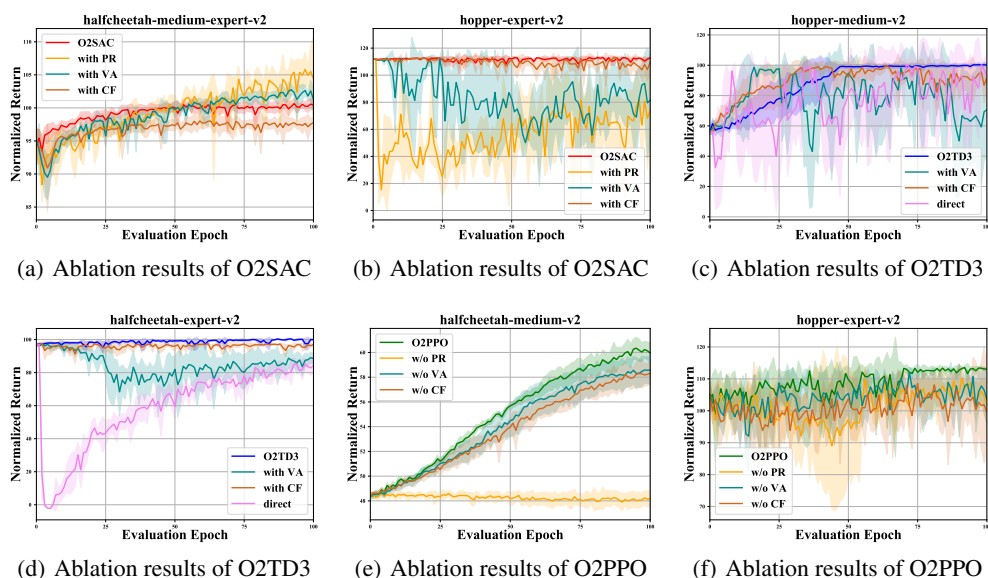

Figure 5: Ablation results of our methods, PR=Policy re-evaluation, VA=Value Alignment, CF=Constrained Fine-tuning. For O2PPO, VA means the use of the auxiliary advantage, and CF means the update of the reference policy.

The results of Figure 5 show that even with a narrow dataset *e.g. hopper-expert-v2*, our methods still achieve stable online fine-tuning. For O2SAC, the fine-tuning process without the optimistic critic reconstruction can lead a sudden performance drop at the beginning phase, e.g. in Figure 5(a), as the offline critic may be severely pessimistic. Due to the alignment of the offline critic and the actor in CQL, using constrained fine-tuning alone can result in overall stable performance improvement,

but O2SAC demonstrates a more efficient result. As for O2TD3, fine-tuning from offline directly suffers from severe performance degradation due to the mismatch of the actor and the critic in offline training, such as in Figure 5(c). Thanks to value alignment, we can address the mismatch, thereby achieving stable performance improvement at the beginning stage. However, due to OOD states during online fine-tuning, constrained fine-tuning is necessary for the long-term stability. At last, PPO is an on-policy method that has a high data quality requirement, so it cannot improve performance if the value is incorrectly evaluated, e.g. in Figure 5(e). Through the reliable auxiliary function, our O2PPO can improve performance stably even with imperfect value evaluation.

Although our methods are universal for any offline algorithm to SAC, TD3 and PPO, some steps can be omitted according to the type of the offline algorithm. For example, for value regularization method *e.g.* CQL, we can only use constrained fine-tuning for stable and efficient O2O RL if the constraint term with a low coefficient has little effect on the critic, as the offline actor and critic are aligned, constrained fine-tuning is enough to combat the problem caused by the drastic jump of Q-values. And for policy constraint methods with explicit constraint *e.g.* TD3+BC and AWAC, as talker above, the critic is evaluated in the same way as online way, so policy re-evaluation can be omitted. For other offline methods with the update way that does not match the online way, such as IQL [20], IVR [47] and $\chi$-QL [13], and methods induced by *behavior-regularized* MDP, and with special policy form *decision transformer*, all steps of our methods are needed for stable O2O RL.

## C  Additional Experiments

### C.1  Difference in policy performance caused by optimistic critic reconstruction

Note that we use $\pi_{on}$ as the initialization of the actor at the beginning of online fine-tuning. In accordance with our analysis, the performance of the initial policy $\pi_{on}$ is expected to align with the characteristics of the actual offline policy. As depicted in Table 3 from our empirical experiments, the results substantiate our analysis, revealing minimal disparities in the performance of $\pi_{on}$ compared to the actual offline policy. In most environments, $\pi_{on}$ even demonstrates superior performance than the offline policy. We attribute this to the effectiveness of our *min* operator, which sensibly tightens the policy distribution.

Table 3: The performance of the offline policy $\pi_{off}$ and the policy $\pi_{on}$ obtained after policy re-evaluation and value alignment

| Dataset | O2SAC | | O2TD3 | |
|---|---|---|---|---|
| | $\pi_{off}$ | $\pi_{on}$ | $\pi_{off}$ | $\pi_{on}$ |
| halfcheetah-medium-v2 | 46.85 | 53.04 | 48.12 | 48.46 |
| hopper-medium-v2 | 59.29 | 65.28 | 56.29 | 61.44 |
| walker2d-medium-v2 | 81.08 | 78.63 | 83.13 | 84.16 |
| halfcheetah-medium-replay-v2 | 45.27 | 48.14 | 44.28 | 45.21 |
| hopper-medium-replay-v2 | 88.97 | 99.64 | 59.26 | 49.95 |
| walker2d-medium-replay-v2 | 80.31 | 75.77 | 77.16 | 75.90 |
| halfcheetah-medium-expert-v2 | 94.14 | 95.43 | 91.99 | 78.64 |
| hopper-medium-expert–v2 | 84.15 | 94.15 | 101.60 | 106.44 |
| walker2d-medium-expert–v2 | 109.44 | 109.30 | 110.51 | 110.53 |
| halfcheetah-medium-expert–v2 | 96.65 | 99.80 | 96.89 | 97.17 |
| hopper-expert–v2 | 111.80 | 111.68 | 111.74 | 106.31 |
| walker2d-expert–v2 | 109.84 | 109.87 | 110.15 | 110.3 |

### C.2  Experiments on D4RL AntMaze navigation tasks

For the difficult tasks of AntMaze navigation, such as *medium* and *large* environments, TD3+BC is almost completely incapable of training a favourable policy [40]. Fine-tuning with the poor

initialization from such a policy helps little and has minor differences with training a policy from scratch, that should be the concern of hybrid learning [4], so we do not consider it.

Here we demonstrate the results of O2SAC and O2PPO on the the difficult tasks of AntMaze navigation, including *antmaze-medium-play-v2*, *antmaze-medium-diverse-v2*, *antmaze-large-play-v2*, *antmaze-large-diverse-v2* environments.

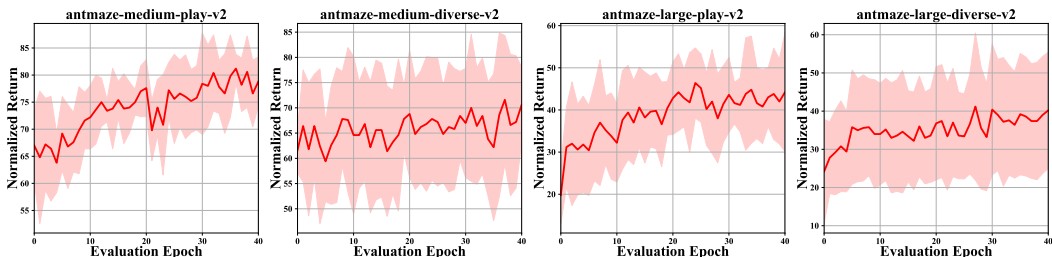

Figure 6: The results of O2SAC on D4RL [9] AntMaze navigation tasks during online fine-tuning. The solid lines and shaded regions represent mean and standard deviation.

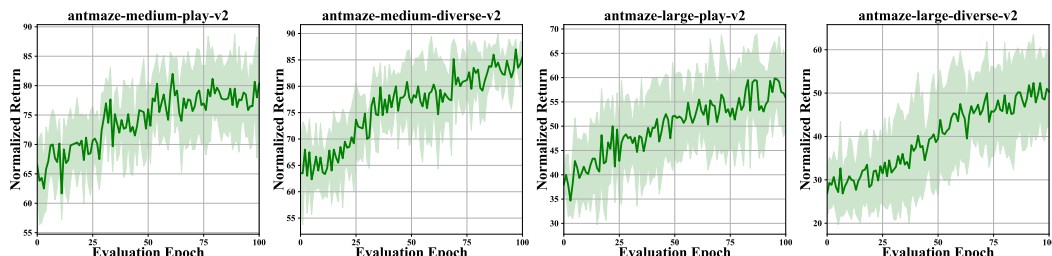

Figure 7: The results of O2PPO on D4RL [9] AntMaze navigation tasks during online fine-tuning. The solid lines and shaded regions represent mean and standard deviation.

Table 4: Average normalized D4RL scores of our methods and some baselines on AntMaze navigation tasks.

| Dataset | AWAC | IQL | PEX | O2SAC | O2PPO |
|---------|------|-----|-----|-------|-------|
| medium-play-v2 | 0.0→ 0.0 | 64.8→78.0 | 73.6→81.0 | 67.0→78.8 | 66.5→80.3 |
| medium-diverse-v2 | 0.0 → 0.0 | 68.8→73.2 | 70.8→83.4 | 61.6→70.6 | 63.5→85.5 |
| large-play-v2 | 0.0 → 0.0 | 39.4→50.2 | 46.2→54.6 | 19.8→44.2 | 38.0→55.83 |
| large-diverse-v2 | 0.0 → 0.0 | 31.0→36.0 | 40.0→58.4 | 24.2→40.2 | 26.83→50.33 |
| total | 0.0 → 0.0 | 204.0→237.4 | 230.6→277.4 | 172.6 →233.8 | 194.8→272.0 |
| Δ | +0.0 | +33.4 | +46.8 | +61.2 | +77.16 |

Note that here we run O2PPO with 250,000 environments steps and run O2SAC with 200,000 environments steps, while in Section 5.1, we run O2PPO with 200,000 environments steps.

Note that as some previous work, in our O2SAC implementation on AntMaze tasks, we do not use the double Q networks trick to avoid overly underestimation, and the threshold $\tau$ reaches the maximum in step 100,000.

Since there is a lack of hyper-parameters for these tasks in many related work [48, 25, 50, 53], we do not compare the results of our methods with other methods. However, according to our simple reproduction, our methods achieve competitive results.

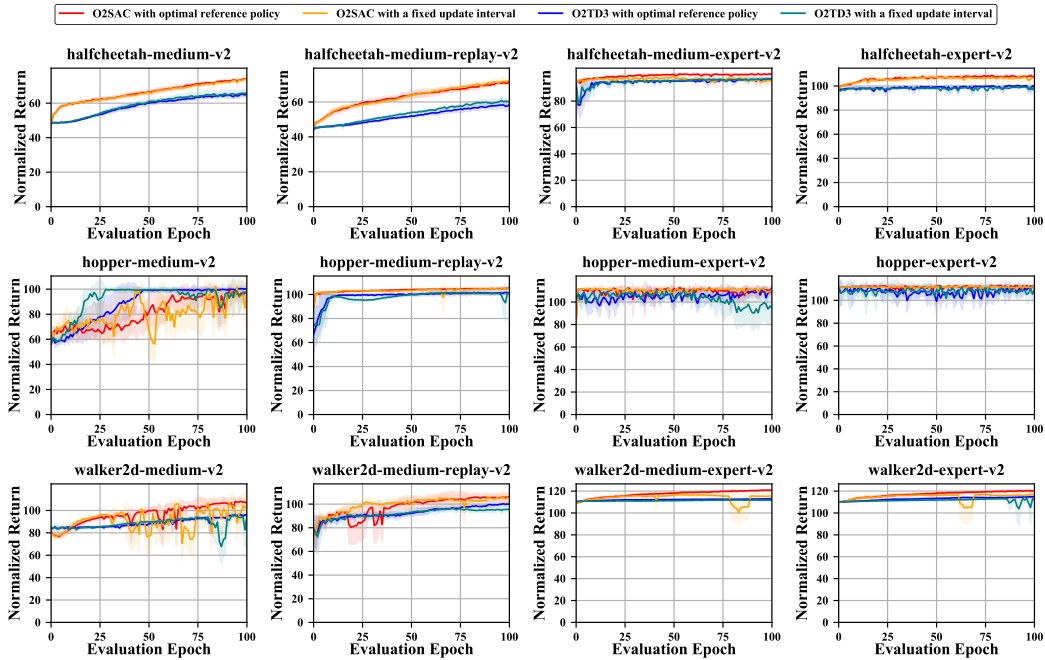

Figure 8: Comparisons on different ways of updating the reference policy for O2SAC and O2TD3.

## C.3 Comparisons with the results of updating the reference policy update at a fixed interval

We set the reference policy as the optimal historical policy during online evaluations in our preceding experiments, and it indeed introduces additional testing information. However, for most of the scenarios that offline RL focus on, this would be acceptable as a well-learned policy can be deployed directly without serious consequence due to its decent performance, e.g. in autonomous driving.

In addition, thanks to the abundant data, another way that does not require practical evaluation is to train a transition model that can be used to evaluate the policy by generating synthetic rollouts as the initial states are provided. Since such a methods is similar to model-based methods, we will leave it for future work.

If evaluation is forbidden, we can replace $\pi_{\mathrm{ref}}$ with a recent policy at a fixed interaction interval. The Corollary 4.3. still holds because the policy performance will be almost certainly improved with a long interaction interval, especially compared to the last reference policy. Considering that the policy performance is most likely to fluctuate in the beginning stage during online fine-tuning, we can set a large update interval for this stage and a small update interval for the subsequent stage, or reduce the threshold change range.

We experiment our methods with a given update interval on D4RL Mujoco locomotion tasks and compare the results with those of the methods with the optimal reference policy, shown as Figure 8 and Figure 9. Without special hyper-parameter optimization, we just update the reference policy per 1000 steps for most environments, but for *hopper-medium-expert-v2* and *hopper-expert-v2*, we update the reference policy per 10000 steps, and as well as for *walker2d-expert-v2* in O2TD3, because these datasets are narrow, making it easy for the policy to suffer from OOD states and actions. With an appropriate interval, the policy performance can achieve competitive improvement when compared with the methods with the optimal reference policy. Policy re-evaluation and Value alignment guarantee the smoothing and stable performance improvement at the beginning stage, and constrained fine-tuning is necessary for the stability of the subsequent stage. Although in *hopper-medium-v2*, O2SAC with a fixed update interval suffers from the performance degradation at the latter stage, because the threshold is large and the Lagrange multiplier $\lambda$ is low, the lack of constraint on the policy update in the reliable region. This phenomenon can easily be amended by reducing the threshold change range.

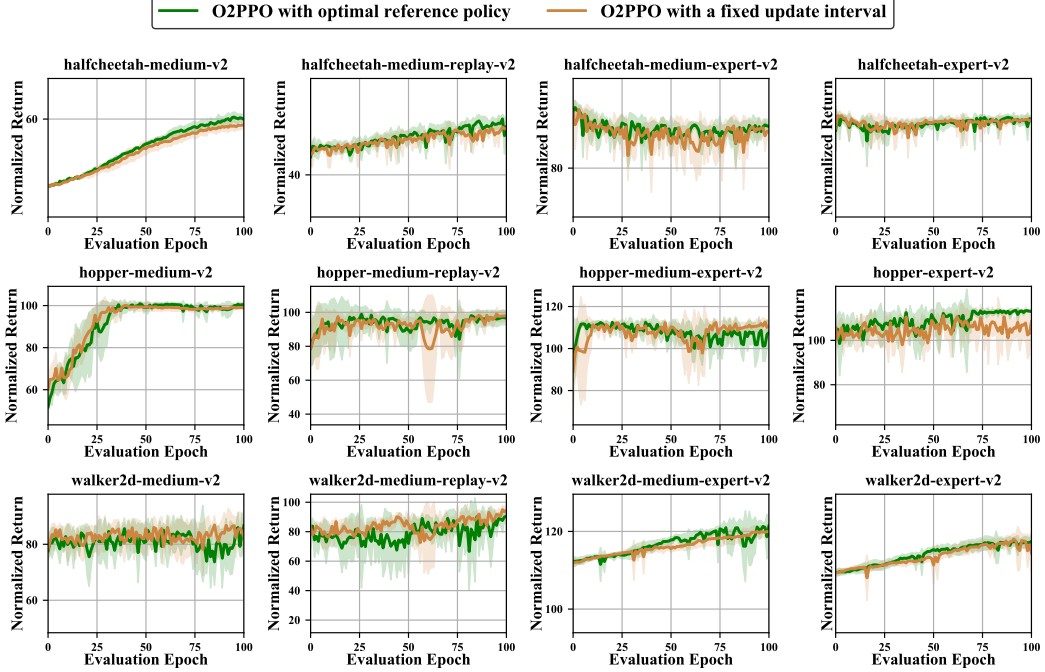

Figure 9: Comparisons on different ways of updating the reference policy for O2PPO.

## C.4 Comparisons with PROTO

In the absence of O2O methods on the deterministic policy, we compare our methods with PROTO [25], a recent O2O method that supports fine-tuning from an offline policy for both the stochastic and the deterministic policy. We reproduce the code of PROTO in Pytorch for the initialization of the policies derived from CQL and TD3+BC, and set all hyper-parameters as in the official paper [25]. As shown in Figure 10, our methods demonstrate significant superiority in both stability and effectiveness, especially for the deterministic policy.

## C.5 Comparisons on initialization of different offline methods

We conducted some experiments for O2SAC with the initialization of different offline methods, including CQL, IQL and ODT. The results are shown in Figure 11. The initial performance of O2SAC initialized from ODT is lower than others since the simple behavior cloning (we directly maximize the likelihood of the actions output by offline ODT while keeping an appropriate entropy) could harm the performance, as discussed in Appendix I. But in hopper-medium-v2, the performance improves quickly. We analyze that by the constraint, the policy can recover the offline performance (about 97 normalized score of ODT), as the output of the cloned policy is near the ODT policy. The results demonstrate that our methods are suitable for any offline algorithm, even the policies are heterogeneous.

## C.6 Accelerate learning with sample-efficient RL

An additional benefit of our methods is their straightforward compatibility with sample-efficient online RL algorithms. Since we only add a constraint that can be considered as part of the reward, the policy iteration process remains consistent with the normal online approach, which makes it feasible to incorporate techniques from advanced efficient RL algorithms. Drawing from [4], we conducted some experiments using a high UTD ratio of 10 (but still update the lagrangian multiplier once per step) and achieved better performance improvements, as shown in Figure 12.

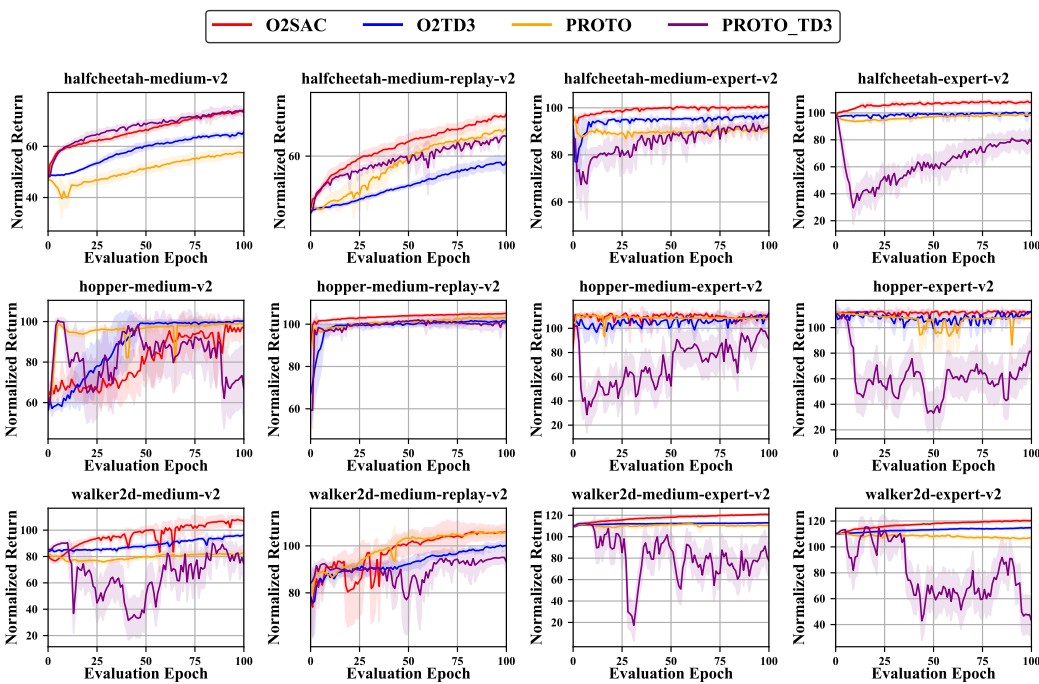

Figure 10: Comparisons with PROTO and PROTO+TD3 [25] on D4RL [9] MuJoCo locomotion tasks during online fine-tuning. The solid lines and shaded regions represent mean and standard deviation.

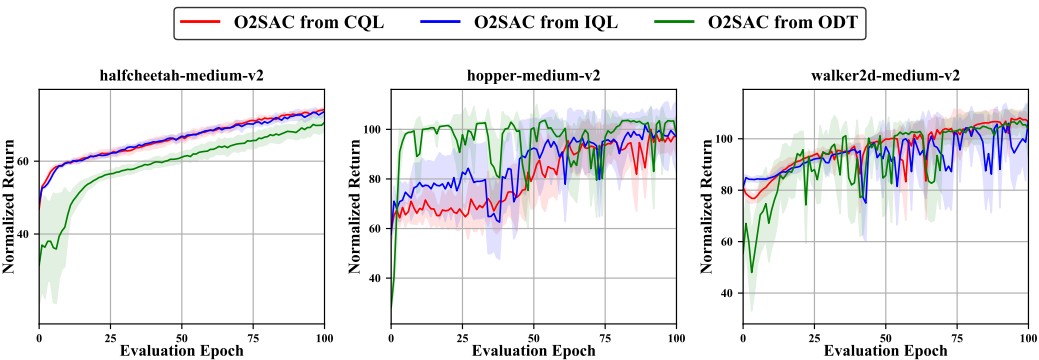

Figure 11: The performance of O2SAC with the initialization from different offline algorithms. The solid lines and shaded regions represent mean and standard deviation.

## C.7   Computational cost for each component

We conducted an experiment on *hopper-medium-v2* environment using the O2SAC method and listed the time cost for different phases in Table 5, evaluated on an Nvidia 3070 GPU.

Table 5: Computational cost for each component of O2SAC

| Training Phase | Offline(CQL) | Policy Re-evaluation | Value alignment |
|---|---|---|---|
| Training Steps | 1M | 0.5M | 0.5M |
| Time Cost | 5.4h | 0.95h | 2.0h |

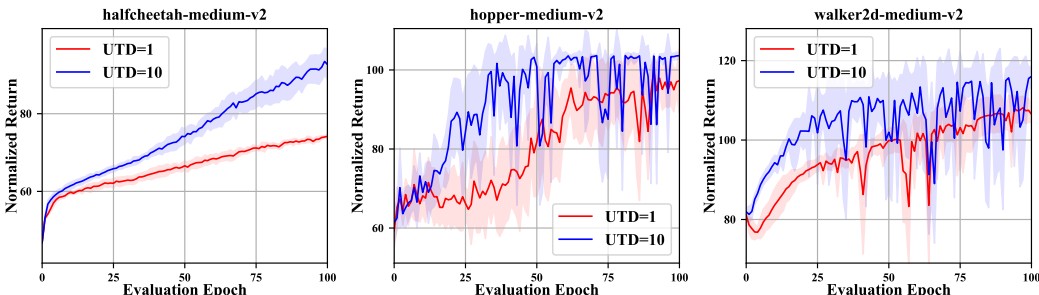

Figure 12: The performance with a high Update-To-Data ratio of O2SAC. The solid lines and shaded regions represent mean and standard deviation.

In policy re-evaluation, since the policy is fixed and the re-evaluation of the critic is straightforward, the computational cost of re-evaluation is significantly lower than that of offline learning. The time cost of value alignment is somewhat higher but still less than that of the offline phase. In fact, the time cost is approximately proportional to the offline phase according to the alignment steps, since both the actor and critic are updated in the value alignment phase.

However, it is worth noting that although we set the training steps for value alignment at 500k, in some environments, only a few alignment steps are needed to calibrate the critic with the offline actor, as shown in Fig. 1 and Fig. 11. Only in antmaze environments, where it is hard for the critic to capture the sparse reward signal, more alignment steps are necessary. Additionally, in constrained fine-tuning, since only the lagrangian multiplier is added to be updated and the interaction cost dominates, the time cost increases very little.

Moreover, in O2O RL, we are typically not concerned about the time cost in the offline process, as different offline methods take different amounts of time. Instead, we prioritize the cost of interactions during online fine-tuning. Our method re-evaluates and aligns the critic with the offline actor solely within the offline dataset, making the time cost less critical.

## D  Related Work

**Unified algorithms for offline-to-online** Many offline algorithms are unified across phases in O2O RL [32, 20, 13, 43, 44], who share the common philosophy of designing an RL algorithm that is suitable for both offline and online phases and then the network parameters trained in the offline phase can be reused for further learning in the online phase. Most of them are policy constraint methods which learn policy without querying OOD samples [20, 43] or penalize action probabilities of them by an explicit estimation of behavior policy [44], which is beneficial for offline performance but limits efficient performance improvements in online fine-tuning, as talked about in [33]. For explicit policy constraint, recent works focus on how to make the constraint adaptive [52, 26] or loosen it gradually [5]. Although such algorithms achieve great performance in offline phase, there is a lack of research on from such algorithms to more efficient online algorithms.

**Efficient online fine-tuning** Some recent works aim to online fine-tune efficiently with optimistic exploration. [15] propose a unified uncertainty-guided framework to explore optimistically in online fine-tuning phase and keep the offline constraint for OOD actions to avoid erroneous update. [30] and [51] utilize ensemble Q-learning to alleviate distribution shift, and implement optimistic exploration by some approaches about ensemble in online RL. For model-based O2O RL, [31] explores regions with high uncertainty and returns in learned model. [42] and [50] concatenate different offline and online algorithms by resetting a new online policy learning from the offline policy gradually, where optimistic exploration is implemented by the new policy. With a given policy and dataset, [3] focus how to recover the performance of the policy rapidly, whose setting is suitable for our method too.

**Stable online fine-tuning** As offline RL works for some important scenarios with high risk, the stable performance is considerable for O2O RL. After [23] put forward distribution shift problem in O2O RL which is alleviated by pessimistic Q-ensemble and balanced replay proposed by them, many methods are proposed to combat this problem. From offline to SAC, [33] mitigates the penalty for

OOD actions in CQL and calibrates Q-values of them by Monte Carlo returns, while [48] reconstructs Q-functions aligned with offline policy. [25] also induces KL divergence to policy objective, which is about current policy and the policy at the last iteration to perform a trust-region-style update, but the performance will drop suddenly at the beginning of online fine-tuning. [19] and [28] utilize environment dynamics model ensemble to obtain uncertainty to penalty OOD actions.

# E    Detailed Discussions about Mismatch in SOTA Offline Algorithms

**CQL**  For common implementation of CQL, [21] do not specify exactly how $\pi_k$ is updated but only provide properties on the policy $\exp(f_k(s,a)/Z(s))$ like SAC. In this way, the actor of CQL is only related to the critic, and the action probability is directly proportional to $Q(s,a)$ like online RL. However, as the conservative policy evaluation operator used to update the Q-function (according to [21] Appendix C, equation 13), the policy is not only related to rewards, but also related to the distribution of behavior policy. The conservative policy evaluation operator in CQL is:

$$Q(s,a) = B^\pi Q(s,a) - \alpha\left[\frac{\mu(a|s)}{\pi_\beta(a|s)} - 1\right] \tag{21}$$

where $B^\pi$ is the standard Bellman operator $B^\pi Q(s,a) = r(s,a) + \gamma\mathbb{E}_{s'\sim P(\cdot|s,a)}[V(s')]$ and

$$V(s) = \mathbb{E}_{a\sim\pi(\cdot|s)}[Q(s,a)] \tag{22}$$

If we define $Q_o$ as the Q-function derived from Eq. (1), which satisfies:

$$Q_o(s,a) = B^\pi Q_o(s,a) \tag{23}$$

Therefore, the relationship of conservative $V(s)$ in CQL and $Q_o(s,a)$ is:

$$V(s) = \mathbb{E}_{a\sim\pi(\cdot|s)}\left[Q_o(s,a) - \alpha\left[\frac{\mu(a|s)}{\pi_\beta(a|s)} - 1\right]\right] \tag{24}$$

The objective of policy is to maximize $V(s)$, so compared with online RL, there is a difference in learning objective due to pessimism. In fact, drawing from SAC, we can derive the objective of CQL:

$$\max_\pi \mathbb{E}\left[\sum_{t=0}^\infty \gamma^t(r(s_t,a_t) - \alpha\left[\frac{\mu(a_t|s_t)}{\pi_\beta(a_t|s_t)} - 1\right])\right] \tag{25}$$

which is in according with *behavior-regularized* MDP proposed by [47] at $f(x) = x - 1$. The conservative policy objective causes pessimistic Q-values estimation, hence in online fine-tuning, the change of policy objective leads to drastic Q-values jump and performance degeneration.

**IQL**  As expectile regression used to estimate expectiles of the state value function with respect to random actions, $Q(s,a)$ and $V(s)$ are not induced by the policy $\pi$, and we denote them as $Q^\mu(s,a)$ and $V^\mu(s)$, where $\mu$ represents a special policy related to the expectile regression [18]. With initialization of such $Q^\mu(s,a)$ and $V^\mu(s)$ in online fine-tuning phase, if we update the actor and the critic in online algorithm, the actor update will tends to $\mu$ at the beginning, which causes unknown performance change because the performance of $\nu$ is unknown. And usually the performance will deteriorate significantly the probability of a satisfactory policy $\nu$ is low.

On the other hand, the work of [47] reveals that the optimal critic of IQL can be derived from *behavior-regularized* MDP at $f(x) = \log(x)$. Meanwhile, the update of actor is derived from return maximization with a KL divergence constraint, it is easily to discovery that the learned policy is relevant to dataset. Therefore, the O2O RL for IQL still suffers from the problem of policy objective change. And compared with value regularization methods, there is not only pessimistic estimation problem but also initial misalignment problem, which means offline actor cannot be derived from offline critic in online update way. In fact, in offline RL, the policy update function is to maximize Eq. (9), which is highly related to behavior policy as $\alpha$ is a fixed hyper-parameter. If 1 is directly used to fine-tune policy, the policy will tends to be only proportional to $Q(s,a)$ of offline critic, hence the policy performance will drop sharply with a great probability.

**TD3+BC**  With a explicit constraint of policy update, the improvement mismatch is obvious: the critic update follows the Bellman backup, but the actor update does not only follow Q-values maximization, that constrains the actor only update in a region close to the dataset. A little different from the

mismatch in IQL, TD3+BC does not suffer from pessimistic estimation since the critic is updated in an optimistic way as the online way, but such improvement mismatch still leads to performance drop. The probability of action with the largest Q-value may be not largest, as talked about in [48], which is not in according with online RL and leads to erroneous update to destroy performance at the beginning of online fine-tuning. Such improvement mismatch exists in almost all explicit constraint methods, but their critics are optimistic because the update ways is the same as online RL, so value alignment is necessary for O2O RL from such offline methods.

# F   Proof

**Proof of Corollary 4.2** The theoretical analysis about FQE can be found in **Theorem 4.2** of [22] or **Theorem 4.9** of [29], here we use the latter result. Full proof is similar to the proof of [29], but a little different from [29], we fix the evaluated policy as the offline policy.

Here we simply show the proof process and emphasis the difference, see [29] for more details.

Since FQE deals with the following regression problem

$$Q_k \leftarrow \underset{Q \in F}{\arg\min} \sum_{i=1}^{|\mathcal{D}|} [Q(s_i, a_i) - r_i - \gamma Q_{k-1}(s_i', \pi(s_i'))]^2, \tag{26}$$

we can have

$$\begin{aligned} |r_i + \gamma Q_{k-1}(s_i', \pi(s_i'))| &\leq 1 + \gamma \bar{V} \leq 2\bar{V} \\ |\mathcal{T}^\pi Q_{k-1}(s, a)| = |r(s, a) + \gamma \mathbb{E}_{s' \sim P(\cdot|s,a)} Q_{k-1}(s', \pi(s'))| &\leq 1 + \gamma \bar{V} \leq 2\bar{V} \end{aligned} \tag{27}$$

With the inherent Bellman evaluation error $d_F^\pi$, we can apply least squares generalization bound here (Lemma A.11 in [2]). With probability at least $1 - \delta$, we have

$$\|Q_k - \mathcal{T}^\pi Q_{k-1}\|_{2,\rho^\beta}^2 \leq \frac{22\bar{V}^2 \log(|\mathcal{F}|/\delta)}{|\mathcal{D}|} + 20d_F^\pi \tag{28}$$

Note that here we fix the policy $\pi$ as the offline policy, that means for a given $Q_0$, $Q_{k-1}$ is well-determined, so we do not need to apply a union bound over all possible $Q_{k-1}$.

Then we first bound $\|Q_k - Q^\pi\|_{2,d_t^\pi \times \pi}$ as the same as [29] did.

$$\begin{aligned} \|Q_k - Q^\pi\|_{2,d_t^\pi \times \pi} &= \|Q_k - \mathcal{T}^\pi Q_{k-1} + \mathcal{T}^\pi Q_{k-1} - Q^\pi\|_{2,d_t^\pi \times \pi} \\ &\leq \|Q_k - \mathcal{T}^\pi Q_{k-1}\|_{2,d_t^\pi \times \pi} + \|\mathcal{T}^\pi Q_{k-1} - Q^\pi\|_{2,d_t^\pi \times \pi} \\ &\leq \sqrt{C}\|Q_k - \mathcal{T}^\pi Q_{k-1}\|_{2,\mu} + \|\mathcal{T}^\pi Q_{k-1} - \mathcal{T}^\pi Q\|_{2,d_t^\pi \times \pi} \text{ (Assumption 4.1)} \\ &= \sqrt{C}\epsilon + \sqrt{\mathbb{E}_{(s,a) \sim d_t^\pi \times \pi}[\gamma \mathbb{E}_{s' \sim P(\cdot|s,a), a' \sim \pi(\cdot|s')}(Q_{k-1}(s', a') - Q^\pi(s', a'))]^2} \\ &\leq \sqrt{C}\epsilon + \gamma\sqrt{\mathbb{E}_{(s,a) \sim d_t^\pi \times \pi, s' \sim P(\cdot|s,a), a' \sim \pi(\cdot|s')}(Q_{k-1}(s', a') - Q^\pi(s', a'))^2} \\ &= \sqrt{C}\epsilon + \gamma\|Q_{k-1} - Q^\pi\|_{2,d_t^\pi \times \pi} \end{aligned} \tag{29}$$

The fifth derivation uses Jensen's inequality. With

$$\|Q_K - Q^\pi\|_{2,d_t^\pi \times \pi} \leq \sum_{k=0}^{K-1} \gamma^k \sqrt{C}\epsilon + \gamma^K \|Q_0 - Q^\pi\|_{2,d_t^\pi \times \pi} \leq \frac{1 - \gamma^K}{1 - \gamma}\sqrt{C}\epsilon + \gamma^K \bar{V} \tag{30}$$

Therefore

$$
\begin{aligned}
|Q^\pi - \hat{Q}^\pi| &= |Q^\pi - Q_K|(K \to \infty) \\
&\leq \|Q_K - Q^\pi\|_{2,d^\pi \times \pi}(\mathrm{Jensen's\ inequality}) \\
&= \sqrt{\mathbb{E}_{d^\pi}\mathbb{E}_{a \sim \pi(\cdot|s)}[Q_K(s,a) - Q^\pi(s,a)]} \\
&= \sqrt{\sum_s (1-\gamma)\sum_{t=0}^\infty \gamma^t d_t^\pi(s) \sum_a \pi(a|s)[Q_K(s,a) - Q^\pi(s,a)]} \\
&= \sqrt{(1-\gamma)\sum_{t=0}^\infty \gamma^t \sum_s d_t^\pi(s) \sum_a \pi(a|s)[Q_K(s,a) - Q^\pi(s,a)]} \\
&= \sqrt{(1-\gamma)\sum_{t=0}^\infty \gamma^t \|Q_K - Q^\pi\|_{2,d_t^\pi \times \pi}^2} \\
&\leq \sqrt{(1-\gamma)\sum_{t=0}^\infty \gamma^t (\frac{1-\gamma^K}{1-\gamma}\sqrt{C}\epsilon + \gamma^K \bar{V})^2} \\
&= \frac{1-\gamma^K}{1-\gamma}\sqrt{C}\epsilon + \gamma^K \bar{V}
\end{aligned}
\tag{31}
$$

**Proof of Proposition 4.3** We define $Q_{\mathrm{fqe}}(s,a)$ as the state-action values for some actions after the process of policy re-evaluation and than the calibrated value, i.e. $Q_{\mathrm{fqe}}(s,a) \leq Q(s,\dot{a}) - \alpha(\log \pi_{\mathrm{off}}(\dot{a}|s) - \log \pi_{\mathrm{off}}(a|s))$. With the *min* operator in Eq. (12), we can divide actions into two categories, *calibrated actions* or *standard actions*. The former refer to those actions with Q-values to be calibrated, and the latter refer to those actions with unchanged Q-values since they are lower than the calibrated values. We denote them as $a_{\mathrm{cal}}$ and $a_{\mathrm{fqe}}$ respectively.

$$
\begin{aligned}
V_{\mathrm{align}}(s) &= \mathbb{E}_{a \sim \pi_{\mathrm{off}}(\cdot|s)}[Q_{\mathrm{align}}(s,a) - \alpha \log \pi_{\mathrm{off}}(a|s)] \\
&= \sum \pi_{\mathrm{off}}(a|s)[Q_{\mathrm{align}}(s,a) - \alpha \log \pi_{\mathrm{off}}(a|s)] \\
&= \sum \pi_{\mathrm{off}}(a_{\mathrm{cal}}|s)[Q_{\mathrm{cal}}(s,a_{\mathrm{cal}})] + \sum \pi_{\mathrm{off}}(a_{\mathrm{fqe}}|s)[Q_{\mathrm{fqe}}(s,a_{\mathrm{fqe}})]+ \\
&\quad \sum \pi_{\mathrm{off}}(a|s)[-\alpha \log \pi_{\mathrm{off}}(a|s)]
\end{aligned}
\tag{32}
$$

Due to the *min* operator, for *standard actions* $a_{\mathrm{fqe}}$, the calibrated Q-values satisfy $Q_{\mathrm{fqe}}(s,a_{\mathrm{fqe}}) \leq Q_{\mathrm{cal}}(s,a_{\mathrm{fqe}})$, therefore

$$
\begin{aligned}
V_{\dot{a}}(s) &= Q(s,\dot{a}) - \alpha \log \pi(a|s) \\
&= \sum \pi_{\mathrm{off}}(a|s)[Q_{\mathrm{cal}}(s,a) - \alpha \log \pi_{\mathrm{off}}(a|s)] \\
&\geq \sum \pi_{\mathrm{off}}(a_{\mathrm{cal}}|s)[Q_{\mathrm{cal}}(s,a_{\mathrm{cal}})] + \sum \pi_{\mathrm{off}}(a_{\mathrm{fqe}}|s)[Q_{\mathrm{fqe}}(s,a_{\mathrm{fqe}})]+ \\
&\quad \sum \pi_{\mathrm{off}}(a|s)[-\alpha \log \pi_{\mathrm{off}}(a|s)] \\
&= V_{\mathrm{align}}(s)
\end{aligned}
\tag{33}
$$

On the other hand, it is easy to see that $Q_{\mathrm{fqe}}(s,a_{\mathrm{fqe}}) - \alpha \log \pi(a_{\mathrm{fqe}}|s) \leq Q_{\mathrm{cal}}(s,a_{\mathrm{cal}}) - \alpha \log \pi(a_{\mathrm{cal}}|s)$, so

$$
\begin{aligned}
V_{\mathrm{fqe}}(s) &= Q_{\mathrm{fqe}}(s,a_{\mathrm{fqe}}) - \alpha \log \pi(a_{\mathrm{fqe}}|s) \\
&\leq \sum \pi_{\mathrm{off}}(a_{\mathrm{fqe}}|s)[Q_{\mathrm{fqe}}(s,a_{\mathrm{fqe}}) - \alpha \log \pi_{\mathrm{off}}(a_{\mathrm{fqe}}|s)]+ \\
&\quad \sum \pi_{\mathrm{off}}(a_{\mathrm{cal}}|s)[Q_{\mathrm{cal}}(s,a_{\mathrm{cal}}) - \alpha \log \pi_{\mathrm{off}}(a_{\mathrm{cal}}|s)] \\
&= V_{\mathrm{align}}(s)
\end{aligned}
\tag{34}
$$

So $V_{\mathrm{fqe}}(s) \leq V_{\mathrm{align}}(s) \leq V_{\dot{a}}(s)$, i.e. Proposition 4.3 is proved.

**Proof of Proposition 4.4** With simplicity, we consider the term of the auxiliary advantage function without probability ratio clipping

$$
\begin{aligned}
- \mathop{\mathbb{E}}_{s \sim R} \mathop{\mathbb{E}}_{a \sim \pi_{\theta_k}(\cdot|s)} \left[ \frac{\pi_\theta}{\pi_{\theta_k}} A_\alpha(s, a) \right] &= - \mathop{\mathbb{E}}_{s \sim R} \mathop{\mathbb{E}}_{a \sim \pi_\theta(\cdot|s)} [A_\alpha(s, a)] \\
&= - \mathop{\mathbb{E}}_{s \sim R} \sum \pi_\theta(a|s) A_\alpha(s, a) \\
&= \mathop{\mathbb{E}}_{s \sim R} [- \sum \pi_\theta(a|s) \log \pi_{\text{ref}}(a|s) - \sum \pi_\theta(a|s) \mathcal{H}(\pi_{\text{ref}}(\cdot|s)] \\
&= \mathop{\mathbb{E}}_{s \sim R} [CELoss(\pi_\theta(\cdot|s), \pi_{\text{ref}}(\cdot|s)) - \mathcal{H}(\pi_{\text{ref}}(\cdot|s)) \sum \pi_\theta(a|s)] \\
&= CELoss(\pi_\theta, \pi_{\text{ref}}) + C
\end{aligned}
\tag{35}
$$

Therefore, the term of the auxiliary advantage function regularizes the policy update near the reference policy, thereby ensuring reliable update even with inaccurate value estimation. And at the beginning of online fine-tuning, the reference policy is initialized from $\pi_{\text{off}}$.

**Proof of Corollary 4.5** Since our constrained fine-tuning method solves a constrained MDP problem by the method in [41], Theorem 2 in [41] still holds in our method, which can be described as *With the constraint satisfied, RCPO algorithm converges almost surely to a fixed point $(\theta^\star(\mu^\star, \lambda^\star), \mu^\star(\lambda^\star), \lambda^\star)$.*

Such theorem guarantees the convergence of RCPO, and with the our proposed constraint, we can utilize contradiction to proof Corollary 4.5.

**Assumption** According to the convergence of RCPO, we can assume the convergent point $(\theta^\star(\mu^\star, \lambda^\star), \mu^\star(\lambda^\star), \lambda^\star)$ with $\lambda^\star \neq 0$.

**Derivation** As the constraint is $\mathbb{E}_\pi [f(\pi(a_t|s_t), \pi_{\text{ref}}(a_t|s_t))] < \alpha$ and $\pi_{\text{ref}}$ is the best one among old policies during online evaluations, $\pi_{\text{ref}} = \pi^\star$ when the algorithm converges, so the constraint term tends to $-\alpha$. According to the update function Eq. (20) of $\lambda$, $\lambda$ tends to be reduced, which is in contradiction to the Assumption. Therefore, $\lambda^\star = 0$ when the algorithm converges.

Meanwhile, when $\lambda^\star = 0$, the constraint is removed, hence the $\pi^\star$ and $Q^\star$ are the same with the optimal results in the MDP without constraint, which means our method converges the optimal point like online RL but keeps stable improvement.

## G  Algorithm Details

### G.1  O2SAC

**the choice of $\alpha$**

Although $\alpha$ is generally smaller than 1 after offline training, as we use the energy policy to align critic with actor, a small $\alpha$ has no influence on the recovery of offline policy but leads to a wide distribution for policy, which means the policy modeled by Gaussian distribution has a large standard deviation, which is harmful for online exploration. If the standard deviation is large, some OOD actions may be taken and dangerous state may arise during online exploration, which dose not suit the setting of high risk scenarios and affects favourable performance. Therefore we choose a suitable $\alpha$ which is smaller than 1 but not excessively small. For Mujoco locomotion tasks, we determine that alpha is 0.2 for the dataset with medium quality, and is 0.5 for the dataset with expert

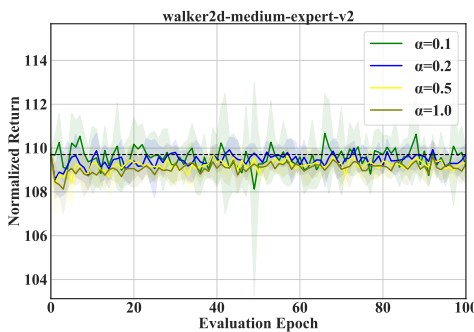

Figure 13: Policy performance during value alignment with different $\alpha$

quality, because small standard deviation induced by a large $\alpha$ is advantageous for online exploration of well trained policy. For AntMaze navigation tasks, we set alpha as 0.5 for *medium* and *large* datasets and 0.2 for *umaze* datasets.

Moreover, as we use *min* operator for value alignment, the limit for $\alpha$ is not so strict. If Q-values induced by policy re-evaluation is smaller than alignment objective, as a Gaussian distribution is used to fit energy policy, such Q-values will leads to a smaller standard deviation.

Note that the choice of $\alpha$ has no effect on the recovery of offline policy, as shown in Figure 13.

**policy re-evaluation** As talked about Section 4.1, with a given $\alpha$, online policy evaluation of SAC Eq. (3) can be used to policy re-evaluation to obtain an optimistic critic.

**value alignment** In Section 4.2, the value alignment method to O2SAC has been described in detail. In brief, we use online policy improvement Eq. (2) to get the actions which have overestimated Q-values, and use Eq. (13) to inhibit their Q-values to a reliable value calculated by maximum entropy RL.

According to the *min* operator, there is no change if Q-values induced by policy re-evaluation is smaller than alignment objective, so our value alignment method not only aligns Q-values with actor but also is as consistent with the results of online evaluation as possible, which reduces Bellman error in online fine-tuning to keep Q-values more stable.

**constrained fine-tuning** For O2SAC, the constraint is KL divergence, and the policy objective of CMDP in O2O RL is:

$$\max \mathbb{E}_\pi[\sum_{t=0}^{\infty} \gamma_t(r_t(s_t, a_t) + \alpha H(\pi(\cdot|s_t)))] \qquad \text{s.t. } \mathbb{E}_\pi[\log(\frac{\pi(a_t|s_t)}{\pi_{\text{ref}}(a_t|s_t)})] < \tau \qquad (36)$$

And corresponding loss functions are:

$$L(\theta) = \max \mathbb{E}_{\pi_\theta}[Q_\mu^{\pi_\theta}(s, a) - \alpha \log \pi_\theta(a|s) - \lambda \log(\frac{\pi_\theta(a|s)}{\pi_{\text{ref}}(a|s)})] \qquad (37)$$

$$L(\mu_i) = \min \mathbb{E}_{(s,a,r,s') \sim R}[(Q_{\mu_i}^{\pi_\theta}(s, a) - y)^2] \qquad (38)$$

$$y = r + \gamma \mathbb{E}_{a' \sim \pi_\theta(\cdot|s')}\left[Q_{\bar{\mu}}^{\pi_\theta}(s', a') - \alpha \log \pi_\theta(a'|s') - \lambda \log(\frac{\pi_\theta(a'|s')}{\pi_{\text{ref}}(a'|s')})\right]$$

$$\begin{aligned} L(\lambda) &= \min_{\lambda \geq 0} - \lambda \mathbb{E}_{s \sim R, a \sim \pi_\theta(\cdot|s)}[\log(\frac{\pi_\theta(a|s)}{\pi_{\text{ref}}(a|s)}) - \tau] \\ &= \min_{\lambda \geq 0} - \lambda \mathbb{E}_{s \sim R, a \sim \pi_\theta(\cdot|s)}[\log \pi_\theta(a|s) - \log \pi_{\text{ref}}(a|s) - \tau] \end{aligned} \qquad (39)$$

A constraint related to offline policy is necessary for online fine-tuning, because OOD states will appear surely during online exploration, especially in narrow dataset, which may lead to erroneous overestimation and destroy old policy. Such constraint can avoid overestimation of actions far away from current policy, hence trust-region style update guarantees stable performance improvement.

Compared with the constraint of current policy and offline policy, our proposed constraint guarantees more optimal update because the policy updated in trust-region of offline policy generally has similar performance to offline policy, which leads to not much performance improvement.

And compared with the tight constraint of current policy and the policy at last iteration, our constraint guarantees rapid recovery when some erroneous update occurs which results in performance degradation, because the Lagrange multiplier $\lambda$ will be larger and make the Q-values of OOD actions lower, hence the policy tends to be close to $\pi_{\text{ref}}$. In such situation, our method can recover a similar performance to $\pi_{\text{ref}}$ for current policy rapidly, but the tight constraint needs to take much time to do that as it constrains the current policy close to the poor policy at last iteration.

### G.2 O2TD3

**policy re-evaluation** Similar to O2SAC, online policy evaluation of TD3 Eq. (6) can be used directly to evaluate an optimistic critic, and in TD3, there is no need for $\alpha$.

**value alignment** As TD3 models a deterministic policy, value alignment method can not be derived directly from the relationship of the actor and the critic like O2SAC. However, from Eq. (5), the gradient of policy is only related to the the gradient of Q(s, a) to a if we fix the policy as offline policy.

So, we have two insights for the actor in TD3: $Q(s, \pi(s))$ is the maximal in $Q(s, \cdot)$ and the gradient around $\pi(s)$ should tend to 0, and the latter also corresponds to the idea of policy smoothing update in TD3.

Note that in most cases, the correlation among different dimensions of action is not be considered, hence we use one-dimensional Gaussian distribution for the following analysis.

Therefore, with the definition that $\dot{a} = \pi(s)$, we consider that normalized Q-values around $\dot{a}$ follow a Gaussian distribution $Q(s,a)/Q(s,\dot{a}) \sim N(\dot{a}, \Sigma)$. And for $Q(s, \dot{a} + \delta)$, we utilize Taylor expansion of first order and omit higher order terms, then we can get:

$$Q(s, \dot{a} + \delta) \approx Q(s, \dot{a}) + \nabla_a Q(s, \dot{a}) \cdot \delta \quad \text{(Taylor expansion)} \tag{40}$$

$$\approx Q(s, \dot{a}) + \nabla_a Q(s, \dot{a} + \delta) \cdot \delta \tag{41}$$

$$= Q(s, \dot{a}) - \frac{\delta^2}{\Sigma} Q(s, \dot{a} + \delta) \tag{42}$$

From Eq. (40) to Eq. (41), we consider the continuous derivative. From Eq. (41) to Eq. (42), we utilize the derivative of a Gaussian distribution $\nabla_x f(x) = -(x - \mu)/\Sigma$, where $f(x)$ is a one-dimensional Gaussian distribution $f(x) \sim N(\mu, \Sigma)$.

Note that $Q(s, \dot{a}) = Q(s, \dot{a})/\sqrt{2\pi\Sigma}$, which means $\Sigma = 1/2\pi$, so we can get:

$$Q(s, \dot{a} + \delta) = \frac{1}{1 + \delta^2/\Sigma} Q(s, \dot{a}) = \frac{1}{1 + 2\pi\delta^2} Q(s, \dot{a}) \tag{43}$$

In our implementation, we replace Eq. (43) to Eq. (15), where $k$ is used to control penalty for overestimated values, and as action range is from -1 to 1, the alignment objective avoids severe underestimation because the minimum Q does not tends to 0 if we set a small $k$, which also reduces Bellman error during online update as the same thing as *min* operator does. And we use the euclidean distance divided by the square root of action dimension to calculate $\delta$, which is $\delta(a, \dot{a}) = \sqrt{\sum (a_i - \dot{a}_i)^2}/|A|$.

Note that in O2TD3, we use the same trick of reward normalization as TD3+BC, which means we subtract 1 from all rewards, so we need to consider the situation of negative rewards. In Eq. (43), we set $Q(s,a)/Q(s,\dot{a}) \sim N(\dot{a}, \Sigma)$ when rewards are positive, so it is natural to set $Q(s,\dot{a})/Q(s,a) \sim N(\dot{a}, \Sigma)$ at the situation of negative rewards. Therefore, for the environment with negative Q-values, we use following way to redress critic:

$$Q(s, \dot{a} + \delta) = (1 + k\delta^2)Q(s, \dot{a}) \tag{44}$$

In a word, we multiply $Q(s, a)$ by $(1 + k\delta^2)$ if $Q(s, a) > 0$, otherwise divide it by $(1 + k\delta^2)$.

**constrained fine-tuning** For O2TD3, the constraint is MSE loss, so the loss functions are:

$$L(\theta) = \max \mathbb{E}_{\pi_\theta}[Q^{\pi_\theta} - \lambda(\pi_\theta(s) - \pi_{\text{ref}}(s))^2] \tag{45}$$

$$L(\mu_i) = \min_{(s,a,r,s') \sim R} \mathbb{E} \left[ \left( Q^{\pi_\theta}_{\mu_i}(s, a) - y \right)^2 \right] \tag{46}$$

$$y = r + \gamma \mathbb{E}_{a' \sim \pi_\theta(\cdot|s')} \left[ Q^{\pi_\theta}_{\bar{\mu}}(s', a') - \lambda(\pi_\theta(s) - \pi_{\text{ref}}(s))^2 \right]$$

$$L(\lambda) = \min_{\lambda \geq 0} - \lambda \mathbb{E}_{s \sim R}[(\pi_\theta(s) - \pi_{\text{ref}}(s))^2 - \tau] \tag{47}$$

### G.3   O2PPO

**policy re-evaluation** Different from O2SAC and O2TD3, as Eq. (8) is not related to offline policy, in practice we can only obtain $V^\mu(s)$ through fitting the returns. Advantages $A^{\pi_{\text{off}}}(s, a)$ computed by such $V^\mu(s)$ may be awfully incorrect, which sequentially causes error update. However, in order to follow the update way of PPO (only $V(s)$ is used to update), we stick to update state value function by fitting the returns, and we introduce a modification to advantages used to update.

**value alignment** As $V^{\pi_{\text{off}}}(s)$ obtained in policy re-evaluation is actually $V^\mu(s)$, which is incorrect to compute advantages $A^\pi(s, a)$ for on-policy update, we propose an auxiliary advantage to redress

erroneous update. Let us think about the property of the auxiliary advantage. First, as an advantage function, its mathematical expectation of offline policy $\pi_{\text{off}}$ should be 0. Second, the function should be able to output positive values and negative values for different actions to distinguish the quality of actions. Last, better actions should correspond higher values. Drawing from the policy form of SAC, we propose the auxiliary advantage as:

$$A_\alpha(s, a) = \alpha \log \pi_{\text{off}}(a|s) + \alpha \mathcal{H}(\pi_{\text{off}}(\cdot|s))$$
$$\mathcal{H}(\pi_{\text{off}}(\cdot|s)) = -\mathbb{E}_{a \sim \pi_{\text{off}}(\cdot|s)}[\log(\pi_{\text{off}}(a|s))] \tag{48}$$

When the policy is modeled as Gaussian distribution, it satisfies: (1) $\mathbb{E}_{a \sim \pi_{\text{off}}(\cdot|s)}[A_\alpha(s, a)] = 0$. (2) $A_\alpha(s, a) > 0$ when $\|a - \mu\|_2 < \sigma$. (3) $A_\alpha(s, a) \propto \log \pi_{\text{off}}(a|s)$. And such properties are in accord with requirements of auxiliary advantage function.

It is notable that here is an implicit assumption that the actions with higher probability for $\pi_{\text{off}}$ are better, and for a well trained offline policy, such assumption should be valid, at least for the beginning of online fine-tuning. With such auxiliary advantage function, we can redress advantages computed by incorrect $V^\pi(s)$ to ensure the stable performance improvement during online fine-tuning, especially for the beginning stage.

**constrained fine-tuning** As the auxiliary advantage function has the ability to constrain policy to update in a reliable region near $\pi_{\text{off}}$, that means $\pi_{\text{ref}} = \pi_{\text{off}}$, we just need to replace $\pi_{\text{ref}}$ as the optimal policy during online evaluations to constrain online fine-tuning. Meanwhile, with the increase of interactions, the critic gradually becomes accurate, approximating $V^\pi$. And on-policy method is naturally stable to improve performance by updating in a reliable region, therefore the constraint factor $\beta$ anneals to 0 from 1 during online fine-tuning.

# H    Implementation Details

## H.1    baseline implementation

**Offline results** For obtaining offline policy, we choose CQL [21], IQL [20] and TD3+BC [10], and we reproduce offline results according to a deep offline RL library CORL [40] https://github.com/tinkoff-ai/CORL, all hyper-parameters are the same with the official implementations. The offline results are used in our methods, Off2On and PROTO.

1. **CQL** We reproduce the results of CQL by the code from CORL https://github.com/tinkoff-ai/CORL/blob/main/algorithms/offline/cql.py.

2. **IQL** Like the implementation of CQL, we reproduce the results of IQL by the code from CORL https://github.com/tinkoff-ai/CORL/blob/main/algorithms/offline/iql.py.

3. **TD3+BC** Similar to above, we reproduce the results of TD3+BC by the code from CORL https://github.com/tinkoff-ai/CORL/blob/main/algorithms/offline/td3_bc.py.

**Online fine-tuning results** For offline-to-online algorithms, for most methods, we reproduce the results according to their official implementations. For AWAC [32], IQL [20] and Cal-QL [33], we reproduce the results by the code from CORL [40]. For ACA [48] and PEX [50], we reproduce the results by the official open-source code https://github.com/ZishunYu/Actor-Critic-Alignment, https://github.com/Haichao-Zhang/PEX.

Although PROTO focus on online fine-tuning, but initialize the policy from other offline algorithms in the official implementation. For the sake of fairness, we rewrite the code according to the paper and official code (https://github.com/Facebear-ljx/PROTO) of PROTO in Pytorch, and we use the offline results of CQL and TD3+BC for initialization respectively.

In addition, in the official implementation of Off2On [23], the policy update 1,000 times per 1,000 environment steps, that is different from the common implementation. Therefore, we reproduce by using all parts that related to the prioritized replay in the official code from https://github.com/shlee94/Off2OnRL, and we use offline results of CQL for initialization with the ensemble size of 5, that is the same as the official paper and implementation.

All hyper-parameters are the same as the paper. And as O2O baseline methods are all off-policy methods, they are allowed to interact with the environment in 100,000 steps.

## H.2 General implementation of our methods

**Network Architecture** As we need to re-evaluate policy, which means we only need offline policy and do not use the offline critic, we can modify the critic network architecture for stable online fine-tuning. In our implementation, we adopt the same network architecture as offline phase but apply Layer Normalization (LayerNorm) for the output of hidden layers after activation function.

[49] indicate that in offline RL, LayerNorm is a good solution to effectively avoid divergence without introducing detrimental bias, leading to superior performance. Moreover, [4] find that LayerNorm is favourable for efficient online RL with offline data. Therefore, for stable evaluation and future consideration, we decide to apply LayerNorm to online critic network.

**Initialization of online replay buffer** We test our method by initializing the online replay buffer with four different types: (1) Initialize the buffer with the entire offline dataset akin to [20]. (2) Conduct a separate online buffer and sample symmetrically from both offline dataset and online buffer akin to [4]. (3) Initialize the buffer with a small number of offline data with high quality akin to [48]. (4) Initialize the buffer without any offline data. For simplicity, we denote them as *All*, *Half*, *Part*, *Null* respectively.

Table 6: Results for different initialization of online replay buffer.

|  | *All* | *Half* | *Part* | *Null* |
|---|---|---|---|---|
| Total scores | 163.08 | 225.54 | 220.26 | 201.63 |

As the results shown in 6, there is little difference among different ways of initialization, except for $all$, as the the quality of the offline dataset may be low. Although the initialization ways of $part$ and $null$ also show the competitive results, however, to be consistent with [4] and future efficient update, we decide to adopt *Half* as our implementation way. It is notable that online policy is still favourable even if we adopt *Null* initialization, thanks to our adaptive constrained fine-tuning.

**loss weight for $\lambda$ update** Since $\lambda$ in Eq. (20) is a variable applied in all states, it may decrease greatly in a batch. In order to considering more about the situation that needs to be constrained, we modify the term about $\lambda$ in the loss Eq. (20) with a weight.

$$L(\lambda) = \min_{\lambda \geq 0} -\lambda \left[ \mathbb{E}_{\pi_\theta} \omega(f(\pi_\theta(a|s), \pi_{\text{ref}}(a|s))) - \tau \right] \tag{49}$$

where $\omega = |0.7 - \mathbb{I}((f(\pi_\theta(a|s), \pi_{\text{ref}}(a|s))) > \tau)|$ gives a large weight to negative coefficients, thereby constraining the abrupt decrease of $\lambda$.

**The choice of the initial value of $\lambda$** As $\lambda$ is the Lagrange multiplier which is adaptive to the constraint, there is no need to design the initial value of $\lambda$ specially. For *medium* and *large* datasets of Antmaze tasks, we set the initial value as 2.0 since the initial performance is low, and we set the initial value as 2.0 for other tasks.

## H.3 O2SAC implementation

**The choice of constraint threshold $\tau$** As we consider the constraint in O2SAC is KL divergence, and we model policy as a squashed Gaussian distribution, we can directly the KL divergence of Gaussian distribution to approximate the constraint. Since $\pi_{\text{ref}}$ is the best one among old policies during online evaluations and it will be close to $\pi_\theta$ with the performance improvement, we can assume that the standard deviation of $\pi_{\text{ref}}$ is the same as the one of $\pi_\theta$. Therefore, according to the KL divergence of Gaussian distribution, we can get:

$$D_{KL}(N(\mu_\theta, \sigma^2), N(\mu_b, \sigma^2)) = \frac{(\mu_\theta - \mu_b)^2}{2\sigma^2} \tag{50}$$

Therefore, we can determine constraint threshold $\tau$ based on the magnitude of change in the mean. For medium dataset, we constrain that $|\mu_\theta - \mu_b| < \sigma$, hence $\tau = 0.5$. However, due to different standard deviations, this threshold may not accurate, and it is an intuitive idea to loosen the constraint at a later stage. So we set $\tau$ as a linearly increasing variable from 0.125 to 2.0 for *medium* and *medium-replay* datasets, which means the allowable range of policy distribution mean is from $\sigma/2$ to

$2\sigma$. And for *medium-expert* and *expert* datasets, we set it from 0.005 to 0.125 for safe update, which means the allowable range of policy distribution mean is from $\sigma/10$ to $\sigma/2$.

**Clipped log likelihood** Due to the limit of precision of Pytorch and the squashed Gaussian policy used in O2SAC, when the output action $a$ is near 1.0, like 0.99999999, the log likelihood $\log \pi(a|s)$ is will be severely low as the value of action will be computed as 1.0, which will occurs when the log likelihood is need to be computed by given a action in value alignment phase and constrained fine-tuning phase. Therefore, for simple calculation, we set the minimum value of log likelihood is -50, a enough small number, which has litter influence on results.

### H.4    O2TD3 implementation

**The choice of constraint threshold** $\tau$ As TD3 models deterministic policy, it is unable to determine the constraint threshold according to the standard deviation. However, when we inspect the interaction process of TD3, we can find that exploration noise is akin to the standard deviation. So we determine $\tau$ from the exploration noise. First we set exploration noise as 0.1 for medium dataset, which is the same as TD3 learning from scratch. And for expert dataset, we set exploration noise as 0.05 because offline policy is well trained in such dataset, and a lower exploration noise is favourable to avoid poor action samples.

After determination of exploration noise, with the idea of the equivalence of the standard deviation and the exploration noise, we set $tau$ similar to O2SAC. For *medium* and *medium-replay* datasets, $tau$ grows linearly from 0.0025 to 0.01, which means the allowable range of policy distribution mean is from $\sigma/2$ to $2\sigma$, when we consider the exploration noise is equal to the standard deviation. Similarly, for *medium-expert* and *expert* datasets, we set it from 0.000025 to 0.000625.

### H.5    O2PPO implementation

We implement our O2PPO basically according to the code of Uni-O4 [24].

**The decay rate of** $\beta$ In our implementation for O2PPO, we set the interaction steps as 250,000 since PPO is an on-policy method which improves performance more slowly than off-policy methods. Since we keep the idea that in *medium-expert* and *expert* dataset, the offline policy is well learned, we decay $\beta$ from 1 to 0 linearly in 500,000 steps, which means in the end of our fine-tuning, $\beta = 0.5$. And for policies trained in other datasets, including AntMaze tasks and *medium* and *medium-replay* datasets of Mujoco tasks, we decay $\beta$ from 1 to 0 linearly during 250,000 steps to reserve more potential of performance improvement.

**Clipped standard deviation** As shown in 14, the policy trained by IQL algorithm has a large standard deviation which makes poor exploration performance, especially in *hopper-medium-expert-v2* and *hopper-expert-v2*. Therefore, we set the maximum value of the standard deviations of policies trained in the two datasets as 0.05, which corresponds to the set of exploration noise in O2TD3, because for a deterministic policy, exploration noise and standard deviation of a stochastic policy are equivalent during taking actions in exploration.

**Shaped and weighted auxiliary advantage** For one-dimensional Gaussian distribution $f = N(\mu, \sigma^2)$, the entropy is $\log(\sqrt{2\pi e \sigma^2})$, which is equal to $\log(f(\mu \pm \sigma))$, hence the maximum value of $A_\alpha(s, a)$ in (48) is $1/2$. However, the minimum value of it may be much low, which leads to unstable training. Therefore, we use SoftPlus activation function to clip the range of $A_\alpha(s, a)$. For each dimensional, we clip $A_\alpha(s, a)$ as follows:

$$\text{Clip}(A_\alpha(s, a)) := \text{SoftPlus}(A_\alpha(s, a) + 4) - 4 \tag{51}$$

By such clipping operator, the range of $A_\alpha(s, a)$ is $(-4, 1/2)$ approximately. Note that when the current policy is close to the reference policy, the actions with overly low logarithmic values are rare, thereby having a little effect on the results.

Note that in the implementation of PPO, the normalization of advantages is needed. In order to make $A_\alpha(s, a)$ and $A(s, a)$ the same order of magnitude, we reweight $A_\alpha(s, a)$ by multiplying a coefficient that is the double value of the standard deviation of $A(s, a)$. Therefore, the maximum value of $A_\alpha(s, a)$ is the standard deviation of the batch of $A(s, a)$.

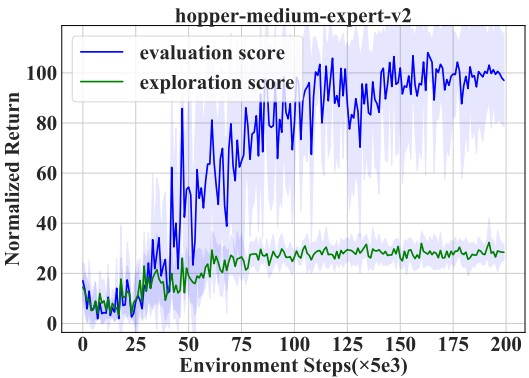

Figure 14: Normalized return of evaluation and exploration during IQL offline training, where evaluation means policy output the mean of action distribution and exploration means actions are sampled from the distribution.

# I   Connect Different Offline and Online RL Methods

Our methods permit connecting different offline and online RL methods as only offline policy is needed in our methods. Therefore, for those RL-based methods, which means the policy is modeled as a stochastic policy or deterministic policy in traditional RL way, it is easy to implement stable online fine-tuning by our methods. For those methods which model policy with different form, such as decision transformer (DT) [6], our methods can be applied easily if a stochastic policy or deterministic policy is used to clone the offline policy.

For behavior cloning of a stochastic policy, the easiest way is to Maximize log likelihood, however, we recommend the form akin to [53], as follows:

$$\min_{\theta} \mathbb{E}_{(s \sim D, a \sim \pi_{\text{off}}(\cdot|s))}[-\log(\pi_\theta(a|s)) - \lambda[\bar{\mathcal{H}}(\pi_\theta(\cdot|s)]] \tag{52}$$

The purpose to maximize the entropy is to avoid an excessively narrow distribution, which may cause drastic jump of Q-values in O2SAC as $\alpha$ will decay to zero quickly. However, in our experiment, such a way of behavior cloning will leads to performance degeneration, which is a question for imitation learning, It is more easy for the behavior cloning of the deterministic policy, MSE loss can be used to update the policy for the states in offline dataset.

It is notable that degree of coverage has an important influence on behavior cloning, which is still an intractable challenge in imitation learning. Our methods provide the opportunity for the application of advanced imitation learning to O2O RL. In addition, even though the policy initialized with imperfect performance, our methods achieve better online performance than the online algorithm learning from scratch.

# J   Pesudo-Codes

**Algorithm 1** O2SAC

**Require:** offline policy $\pi_{\text{off}}$, offline dataset $D$, factor $\alpha$, threshold $\tau$, interaction interval T
   *# Optimistic critic reconstruction before online fine-tuning*
1: Initialize the actor $\pi_{\text{on}}$ with $\pi_{\text{off}}$ and the critic $Q_{\text{on}}$ with random parameters
2: **for** iteration $i = 1, 2, \cdots$ **do**
3:    Update the critic in the optimistic way by 3                                    ▷ Policy re-evaluation
4: **end for**
5: **for** iteration $i = 1, 2, \cdots$ **do**
6:    Update $\pi_{\text{on}}$ to seek overestimated actions by 2
7:    Update $Q_{\text{on}}$ to suppress overestimated Q-values by 13                  ▷ Value Alignment
8: **end for**
   *# Constrained online fine-tuning*
9: Set the reference policy $\pi_{\text{ref}}$ as $\pi_{\text{on}}$ and initialize the replay buffer $R$ with $D$
10: **for** iteration $i = 1, 2, \cdots$ **do**
11:    Interact with the environment and store the transition in replay buffer $R$
12:    Update the temperature coefficient $\alpha$ by 4                              ▷ Constrained Fine-tuning
13:    Update the Lagrange multiplier $\lambda$ by 39
14:    Update the critic $Q_{\text{on}}$ by 38
15:    Update the policy $\pi_{\text{on}}$ by37
16:    **if** evaluation permitted **then**
17:       **if** i % interaction interval == 0 **then**
18:          **if** $J(\pi_{\text{on}}) > J(\pi_{\text{ref}})$ **then**
19:             Replace $\pi_{\text{ref}}$ as $\pi_{\text{on}}$
20:          **end if**
21:       **end if**
22:    **else**
23:       Replace $\pi_{\text{ref}}$ as $\pi_{\text{on}}$ at a given interval
24:    **end if**
25: **end for**

---

**Algorithm 2** O2TD3

**Require:** offline policy $\pi_{\text{off}}$, offline dataset $D$, factor $k$, threshold $\tau$, interaction interval
   *# Optimistic critic reconstruction before online fine-tuning*
1: Initialize the actor $\pi_{\text{on}}$ with $\pi_{\text{off}}$ and the critic $Q_{\text{on}}$ with random parameters
2: **for** iteration $i = 1, 2, \cdots$ **do**
3:    Update the critic in the optimistic way by 6                                    ▷ Policy re-evaluation
4: **end for**
5: **for** iteration $i = 1, 2, \cdots$ **do**
6:    Update $\pi_{\text{on}}$ to seek overestimated actions by 5
7:    Update $Q_{\text{on}}$ to suppress overestimated Q-values by 16                  ▷ Value Alignment
8: **end for**
   *# Constrained online fine-tuning*
9: Set the reference policy $\pi_{\text{ref}}$ as $\pi_{\text{on}}$ and initialize the replay buffer $R$ with $D$
10: **for** iteration $i = 1, 2, \cdots$ **do**
11:    Interact with the environment and store the transition in replay buffer $R$
12:    Update the Lagrange multiplier $\lambda$ by 47                                ▷ Constrained Fine-tuning
13:    Update the critic $Q_{\text{on}}$ by 46
14:    Update the policy $\pi_{\text{on}}$ by 46
15:    **if** evaluation permitted **then**
16:       **if** i % interaction interval == 0 **then**
17:          **if** $J(\pi_{\text{on}}) > J(\pi_{\text{ref}})$ **then**
18:             Replace $\pi_{\text{ref}}$ as $\pi_{\text{on}}$
19:          **end if**
20:       **end if**
21:    **else**
22:       Replace $\pi_{\text{ref}}$ as $\pi_{\text{on}}$ at a given interval
23:    **end if**
24: **end for**

---
**Algorithm 3** O2PPO
---
**Require:** offline policy $\pi_{\text{off}}$, offline dataset $D$, factor $\alpha$, interaction interval, update interval
    *# Optimistic critic reconstruction before online fine-tuning*
1: Initialize the actor $\pi_{\text{on}}$ with $\pi_{\text{off}}$ and the critic $V_{on}$ with random parameters
2: **for** iteration $i = 1, 2, \cdots$ **do**                                   ▷ Policy re-evaluation
3:     Update the critic $V_{on}$ in the optimistic way by fitting the returns of offline trajectories
4: **end for**
    *# Value Alignment & Constrained online fine-tuning*
5: Set the reference policy $\pi_{\text{ref}}$ as $\pi_{\text{on}}$ and initialize an empty replay buffer $R$
6: **for** iteration $i = 1, 2, \cdots$ **do**
7:     Interact with the environment and store the transition in replay buffer
8:     **if** i % update interval == 0 **then**
9:         **for** iteration $i = 1, 2, \cdots, K$ **do**        ▷ Value Alignment & Constrained Fine-tuning
10:            Compute advantage by 18
11:            Update the policy $\pi_{\text{on}}$ by 7
12:            Update the critic $V_{on}$ by 8
13:         **end for**
14:         Reset $R$ to empty
15:     **end if**
16:     **if** evaluation permitted **then**
17:         **if** i % interaction interval == 0 **then**
18:            **if** $J(\pi_{\text{on}}) > J(\pi_{\text{ref}})$ **then**
19:                Replace $\pi_{\text{ref}}$ as $\pi_{\text{on}}$
20:            **end if**
21:         **end if**
22:     **else**
23:         Replace $\pi_{\text{ref}}$ as $\pi_{\text{on}}$ at a given interval
24:     **end if**
25: **end for**
---

