# OpenReview forum: "Optimistic Critic Reconstruction and Constrained Fine-Tuning for General Offline-to-Online RL"
_NeurIPS.cc/2024/Conference — NeurIPS 2024 poster_

### Official Review · Reviewer_aiGQ · 2024-07-07

**Soundness:** 4
**Presentation:** 3
**Contribution:** 3
**Rating:** 7
**Confidence:** 4

**Summary:**

The paper proposes a general offline-to-online (O2O) reinforcement learning method that can work with any offline RL algorithm. It addresses evaluation and improvement mismatches between offline datasets and online environments by (1) Re-evaluating the offline critic optimistically; (2) Calibrating the critic with the offline actor; (3) Performing constrained online fine-tuning. This approach shows stable and efficient performance improvements across various simulated tasks compared to existing methods.

**Strengths:**

1. This paper is well-written and easy to follow.
2/ The re-evaluation and calibration procedures are indeed very crucial for improving offline-to-online RL training, the authors make a great effort to show the importance of such procedure, with both empirical evidence and theoretical guarantees.

**Weaknesses:**

There are several minor presentation issues, not very critical, but they significantly affect the visual quality of the paper.
1. In line 86, equation 2, the subscription under the two expectations is getting too close, making the formula a bit messy.
2. In line 132, equation 9, the parenthesis of the first $f$ should be larger, so that it covers the inputs.
3. In figure 3, the colors of some curves are very similar, making it hard to tell the performance of each method.

**Questions:**

N/A

---

> ### Author Rebuttal · Authors · 2024-08-07
>
> Thanks you for the high praise and the comprehensive review of our paper.
>
> **Q1:** There are several minor presentation issues, not very critical, but they significantly affect the visual quality of the paper.
>
> **A1:** Thanks for your suggestions. We have revised these language mistakes. We will proofread the paper and revise writing issues carefully.
>
> Thank you for your review again and we appreciate your suggestions for a better presentation. We are always willing to answer any of your further concerns.

---

> > ### Comment · Reviewer_aiGQ · 2024-08-07
> >
> > Thank you for the feedback, I have no further questions and I would like to keep my rating.

---

### Official Review · Reviewer_9DG2 · 2024-07-15

**Soundness:** 3
**Presentation:** 3
**Contribution:** 3
**Rating:** 6
**Confidence:** 4

**Summary:**

The paper addresses the offline-to-online (O2O) reinforcement learning problem with the goal of improving online performance by leveraging offline data. The primary contributions of this paper are twofold. First, it identifies and elaborates on two key challenges in O2O RL: evaluation and improvement mismatches, which differentiate offline and online RL. Second, it introduces a general method for transferring knowledge from any offline approach to three representative online methods. Both theoretical analysis and empirical experiments thoroughly validate the effectiveness of the proposed method.

**Strengths:**

Significance: In contrast to existing works, this paper is the first to summarize two mismatches between offline and online methods. These mismatches, which relate to two types of offline approaches, reveal their negative effects on subsequent online fine-tuning.

Contribution: The proposed method effectively balances specificity and flexibility. To address the two distinct mismatches, the paper introduces policy re-evaluation and value alignment techniques, which yield optimistic Q-value estimates and accurate Q-value calibration, respectively. Furthermore, the method is applicable to a broad range of representative online methods, demonstrating its wide applicability.

Soundness: The experiments are thorough and strongly support the claims of the paper, including the motivation and effectiveness. Compared to multiple SOTA methods, the proposed method shows superiority.

**Weaknesses:**

The method consists of three components. The first two components are developed to solve two mismatches. How about the third component? Is there another key issue in O2O learning tasks, such as another mismatch?

In value alignment, regarding different online methods, the paper develops different strategies. Does this imply that the method lacks generalizability?

**Questions:**

Although the method appears straightforward, it comprises three components. What is its time complexity? The paper is suggested to include an analysis on time complexity.

**Limitations:**

The paper does not have a Limitation section.

---

> ### Author Rebuttal · Authors · 2024-08-07
>
> Thanks for your high praise and comprehensive review of our paper. We appreciate the questions you raised and are committed to delivering a comprehensive response to address the issues.
>
> **Q1:** The method consists of three components. The first two components are developed to solve two mismatches. How about the third component? Is there another key issue in O2O learning tasks, such as another mismatch?
>
> **A1:** There is indeed another key issue in O2O RL, known as _distribution shift_ in previous work [1][2][3]. This issue arises from the discrepancy between the offline dataset and the interactive data collected by the offline policy, which can negatively impact performance improvement. This challenge occurs whether the policy is evaluated using suboptimal offline data or high-quality but limited data. Although we strive to maintain an optimistic property for the critic and align the critic with the actor, achieving stable online fine-tuning remains challenging due to the inevitable _distribution shift_.
>
> Most current offline algorithms focus on avoiding OOD actions and training a reliable policy on the states present in the dataset. However, due to the inherent optimism in online RL, encountering OOD states and actions is unavoidable, leading to performance fluctuations. This is particularly problematic in critical scenarios, especially high-risk ones. For OOD states, even if the policy is well-trained during the offline phase, it may still fail to produce favorable actions, potentially causing erroneous policy updates.
>
> To conclude, while optimistic critic reconstruction can guarantee stable and efficient performance improvement initially, it is essential to implement constrained fine-tuning in later stages to maintain continued stability.
>
> **Q2:** In value alignment, regarding different online methods, the paper develops different strategies. Does this imply that the method lacks generalizability?
>
> **A2:** It is important to note that the different strategies correspond to different policy types and update mechanisms, as we study the problems of O2O settings from the perspective of online RL. In online RL, the alignment ways between the critic and the actor can vary significantly due to different foundational designs.
>
> O2SAC is designed for stochastic policies updated in an off-policy manner; O2TD3 is for deterministic policies updated in an off-policy manner; and O2PPO is for stochastic policies updated in an on-policy manner. These three methods are representative and cover the major categories of existing mainstream online algorithms, making it straightforward to incorporate other advanced techniques. Thus, rather than indicating a lack of generalizability, the developed strategies demonstrate our method's adaptability to a wide range of policy types and update mechanisms in online RL.
>
>
> **Q3:** Although the method appears straightforward, it comprises three components. What is its time complexity? The paper is suggested to include an analysis on time complexity.
>
> **A3:** In policy re-evaluation, since the policy is fixed and the re-evaluation of the critic is straightforward, the computational cost of re-evaluation is significantly lower than that of offline learning. The time cost of value alignment is somewhat higher but still less than that of the offline phase. In fact, the time cost is approximately proportional to the offline phase according to the alignment steps, since both the actor and critic are updated in the value alignment phase.
> We conducted an experiment on the _hopper-medium-v2_ environment using the O2SAC method and listed the time cost in different phases as follows.
>
> | Trainning Phase | Offline(CQL)| Policy Re-evaluation | Value alignment|
> | -----------     | ----------- | -----------          | -----------    |
> | Trainning Steps | 1M          | 0.5M                 |  0.5M          |
> | Time Cost       | 5.4h        | 0.95h                |  2.0h          |
>
> However, it is worth noting that although we set the training steps for value alignment at 500k, in some environments, only a few alignment steps are needed to calibrate the critic with the offline actor, as shown in Fig. 1 and Fig. 11. Only in _antmaze_ environments, where it is hard for the critic to capture the sparse reward signal, more alignment steps are necessary. Additionally, in constrained fine-tuning, since only the lagrangian multiplier is added to be updated and the interaction cost dominates, the time cost increases very little.
>
> Moreover, in O2O RL, we are typically not concerned about the time cost in the offline process, as different offline methods take different amounts of time. Instead, we prioritize the cost of interactions during online fine-tuning. Our method re-evaluates and aligns the critic with the offline actor solely within the offline dataset, making the time cost less critical.
>
> While the time cost is not the main concern, we will include a detailed analysis of the time complexity in the revised paper to provide a clearer understanding of the computational requirements for each component.
>
> [1] Lee et al. Offline-to-online reinforcement learning via balanced replay and pessimistic q-ensemble, CoRL 2022.
>
> [2] Yu et al. Actor-critic alignment for offline-to-online reinforcement learning, ICML 2023.
>
> [3] Nakamoto et al. Cal-ql: Calibrated offline rl pre-training for efficient online
> fine-tuning, NeurIPS 2023.
>
> Thank you for your constructive review again. We hope we have resolved your concerns. We are always willing to answer any of your further concerns.

---

### Official Review · Reviewer_zZK3 · 2024-07-15

**Soundness:** 3
**Presentation:** 3
**Contribution:** 3
**Rating:** 6
**Confidence:** 1

**Summary:**

This paper proposes to handle evaluation and improvement mismatches in offline-to-online RL. To this end, the authors suggest to re-evaluate the pessimistic critic and calibrate the misaligned critic with the reliable offline actor. Then, they perform constrained fine-tuning. They evaluate the performance in the standard offline-to-online RL benchmark.

**Strengths:**

- The framework is theoretically analyzed.
- Experimental evaluation is extensive.

**Weaknesses:**

- Difficult to follow up on technical novelties. I suggest adding concept figures or algorithm tables to highlight core contributions.

**Questions:**

- Does re-evaluation require heavy computation? Is it more efficient compared to RLPD [1], which initialize replay buffer with offline dataset instead of offline pre-training?

[1] Ball, Philip J., et al. "Efficient online reinforcement learning with offline data." ICML 2023

**Limitations:**

See Weaknesses and Questions.

---

> ### Author Rebuttal · Authors · 2024-08-07
>
> Thank you for your appreciation of our paper. We are glad that you consider our work “theoretical analyzed”. We are glad to answer all your questions.
>
> **Q1:**   Difficult to follow up on technical novelties. I suggest adding concept figures or algorithm tables to highlight core contributions.
>
> **A1:** Thanks for your valuable suggestion. To facilitate understanding, I would like to elucidate the techniques simply again. As the analysis in our paper, in O2O scenarios, evaluation and improvement mismatches are common. Recognizing that the offline policy is well-trained and trustworthy, we first utilize FQE to re-evaluate the critic optimistically, as in the online process. Given factors like partial data coverage, we then calibrate the critic with the offline actor to achieve an optimistic and reliable critic. Finally, to address distribution shift, we incorporate CMDP into online fine-tuning for stable and efficient performance improvement. We have included the pseudocodes in Appendix J.
>
> **Q2:**  Does re-evaluation require heavy computation? Is it more efficient compared to RLPD [1], which initialize replay buffer with offline dataset instead of offline pre-training?
>
> **A2:** Since the policy is fixed and the re-evaluation of the critic is straightforward, the computational cost of re-evaluation is significantly lower than that of offline learning. In our experiments, the re-evaluation takes about 1 hour for O2SAC (whereas CQL takes more than 5 hours) and 40 minutes for O2TD3. For O2PPO, updating the critic by fitting the returns independently of the policy results in even lower time costs.
>
> Generally, in O2O RL, we are typically not concerned about the time cost in the offline process, as different offline methods take different amounts of time. Instead, we prioritize the cost of interactions during online fine-tuning. Our method re-evaluates and aligns the critic with the offline actor solely within the offline dataset, making the time cost less critical.
>
> Moreover, it is challenging to directly compare our method with RLPD because they address different objectives. RLPD focuses on learning a policy from scratch using a given dataset, whereas our method aims to improve a well-trained policy in O2O scenarios with limited interactions in a stable and efficient manner. Nonetheless, we acknowledge that integrating RLPD techniques could potentially enhance our method's performance, as our approach imposes minimal additional constraints during the online process. This is an interesting future direction of our work.
>
> Thank you for your review again. We hope we have resolved your concerns. We are always willing to answer any of your further concerns.

---

### Official Review · Reviewer_QxHN · 2024-07-19

**Soundness:** 3
**Presentation:** 3
**Contribution:** 3
**Rating:** 6
**Confidence:** 4

**Summary:**

The paper proposes a general framework to bridge offline-to-online RL. It first studies two types of issues in O2O RL: evaluation mismatch and improvement mismatch. The proposed method addresses these issues by combining policy re-evaluation, value alignment, and constrained online fine-tuning. Unlike prior methods, the proposed framework can be applied to any offline RL algorithm. In the experiment, the method was built on top of CQL/SAC, TD3+BC, and IQL/PPO, demonstrating its performance in D4RL tasks.

**Strengths:**

1. The breakdown of the two mismatch problems is interesting.

2. The proposed method is compelling in that it can generally bridge any offline RL and online RL algorithms.

3. The results in D4RL locomotion tasks are solid.

4. Overall, the paper is well-written and contains informative ablation studies.

**Weaknesses:**

1. [Major] It is a bit confusing to me which offline pre-training methods were used for each result. Is it correct that all the results from O2SAC are pre-trained with CQL, O2TD3 is pre-trained with TD3+BC, and O2PPO is pre-trained with IQL? I suggest mentioning these details more clearly in the paper.

2. [Major] While the paper suggests that the proposed method is universal to the offline RL algorithm, there is not much comparison of using the same online RL algorithm with different offline pre-training methods. Can you provide ablations of running O2SAC on the same task with different offline methods, such as CQL, IQL, and ODT?

3. [Major] While the results for the harder AntMaze tasks (antmaze-medium/large) are briefly mentioned in the appendix, could the authors provide the full comparisons to previous methods and add them to Table 1?

4. [Major] Is it possible to include results on the Adroit binary task, as in [1, 2, 3], which is a common benchmark for studying the sample efficiency of online RL with offline data?

5. [Minor] Figure 5 (a) is a bit confusing to me. Which plot corresponds to the unconstrained fine-tuning mentioned below?
>(l.550) For O2SAC, unconstrained fine-tuning suffers from a performance fluctuation, and direct fine-tuning from offline may lead to faster performance improvement, but drops sharply in the subsequent phase, e.g. in Figure 5(a)

6. [Minor] For the value alignment objective in Eq.13, can the authors explain more on why L_retain is needed? What will happen if we only use L_align as the objective?

7. [Minor] It would be interesting to see if the proposed method can be combined into a recent sample-efficient online RL algorithm that can use a high UTD ratio, such as RLPD [3].

8.  [Minor] The legend for Figure 3 (c) seems to be incorrect: O2TD3 -> O2PPO

[1] Nair et al., AWAC: Accelerating Online Reinforcement Learning with Offline Datasets, 2020

[2] Nakamoto et al. Cal-QL: Calibrated Offline RL Pre-Training for Efficient Online Fine-Tuning, 2023

[3] Ball et al., Efficient Online Reinforcement Learning with Offline Data, 2023

**Questions:**

The questions are included in the previous section.

**Limitations:**

The limitations are addressed.

---

> ### Author Rebuttal · Authors · 2024-08-07
>
> Thanks for your insightful review and positive recognition of our paper. We are glad that you consider our work “interesting, solid, well-written”. We appreciate the questions you raised and are committed to delivering a comprehensive response to address the issues.
>
> **Q1:** Suggest mentioning the offline pretrained algorithms used for O2SAC, O2TD3 and O2PPO more clearly in the paper.
>
> **A1:** Thanks for your valuable suggestion. We adopt suitable offline algorithm according to the consistency of the policy form just for simple experiments. For example, both CQL and O2SAC adopt the squashed Gaussian policy. In fact, any offline algorithm can be used for the initialization of our methods with a change of policy form through behavior cloning, as discussed in Appendix I of our paper. And we will mention the offline pretrained algorithms used for O2SAC, O2TD3 and O2PPO more clearly in the paper.
>
> **Q2:** While the paper suggests that the proposed method is universal to the offline RL algorithm, there is not much comparison of using the same online RL algorithm with different offline pre-training methods. Can you provide ablations of running O2SAC on the same task with different offline methods, such as CQL, IQL, and ODT?
>
> **A2:** Thanks for your suggestion. We provide the results in Fig. 1 of the uploaded PDF. The initial performance of O2SAC initialized from ODT is lower than others since the simple behavior cloning (we directly maximize the likelihood of the actions output by offline ODT while keeping an appropriate entropy) could harm the performance, as discussed in Appendix I. But in _hopper-medium-v2_, the performance improves quickly. We analyze that by the constraint, the policy can recover the offline performance (about 97 normalized score of ODT), as the output of the cloned policy is near the ODT policy. We will include these results and discussion in the revised version.
>
> **Q3:** While the results for the harder AntMaze tasks (antmaze-medium/large) are briefly mentioned in the appendix, could the authors provide the full comparisons to previous methods and add them to Table 1?
>
> **A3:** Since some work does not give the hyper-parameters of these tasks and TD3+BC performs extremely poor (almost 0) on these tasks resulting in the terrible initialization for O2TD3, it may be inappropriate to add the comparisons to Table 1.  However, we did some comparisons shown in Table 1 of the uploaded PDF. We are willing to take more time to compare with more algorithms like ODT and give a separate table in the revised paper.
>
> **Q4:**   Is it possible to include results on the Adroit binary task, as in [1, 2, 3], which is a common benchmark for studying the sample efficiency of online RL with offline data?
>
> **A4:** We tried the experiments on Adroit binary tasks but found some difficulties of getting a favourable offline policy by CQL and TD3+BC since the corresponding hyper-parameters are not given in the papers. We tried the hyper-parameters of antmaze tasks but achieve extremely poor performance (almost 0). And we tried cloning a policy by the offline policy learned by IQL, but the performance of the cloned policy is still poor. We cannot provide the corresponding results at the moment for O2SAC and O2TD3. In Fig. 2 of the uploaded PDF, we show the results of O2PPO initialized by IQL. Since the Adroit binary tasks return the sparse rewards like antmaze, given the effectiveness of antmaze tasks, it is reasonable to think that our methods can be applied to the Adroit binary tasks.
>
> **Q5:**   Figure 5 (a) is a bit confusing to me. Which plot corresponds to the unconstrained fine-tuning mentioned below?
>
> **A5:** We apologize for the confusion caused by the mismatched expression. The expression refers to plots in a previous version of our paper, but we did not update the expression after revising the paper. Thanks for pointing this out and we will correct it in the revised paper.
>
> **Q6:**   For the value alignment objective in Eq.13, can the authors explain more on why L_retain is needed? What will happen if we only use L_align as the objective?
>
> **A6:** The role of L_align is to suppress the Q-values of OOD actions. At the beginning of value alignment, the target Q-values of OOD actions can be extremely low as $log \pi_{off}(a_{ood}|s)$ can be extremely low, resulting in an overall or even catastrophic underestimation of Q-values, thereby destroying the optimistic property after policy re-evaluation. The role of L_retain is to keep the optimistic property by keeping the Q-values of reliable actions $\dot{a}$, that is necessary for value alignment since we take them as the anchors to calibrate the Q-values.
>
> **Q7:**   It would be interesting to see if the proposed method can be combined into a recent sample-efficient online RL algorithm that can use a high UTD ratio, such as RLPD [3].
>
> **A7:** Yes, this is indeed another advantage of our method. Since we only add a constraint that can be considered as part of the reward, the policy iteration process remains consistent with the normal online approach, which makes it feasible to incorporate techniques from advanced efficient RL algorithms. We conducted some experiments using a high UTD ratio of 10 (but still update the lagrangian multiplier once per step) and achieved better performance improvement, as shown in Fig. 3 of the uploaded PDF. We will include these results and discussions in the revised version.
>
> **Q8:**   The legend for Figure 3 (c) seems to be incorrect: O2TD3 -> O2PPO
>
> **A8:** Thanks for your careful review. We will correct this mistake in our revised paper.
>
> Thank you for your insightful review again. We hope we have resolved your concerns. We are always willing to answer any of your further concerns.

---

> > ### Comment · Reviewer_QxHN · 2024-08-13
> >
> > I thank the authors for the additional results and clarifications. Overall, I find my concerns have been addressed. I will increase the score to 6.

---

### Author Rebuttal · Authors · 2024-08-07

Thank you to all the reviewers for your thorough evaluation of our paper. Your constructive comments have been invaluable in helping us enhance our work.

---

### Decision · Program_Chairs · 2024-09-25

**Decision:**

Accept (poster)

**Comment:**

This paper analyzes two issues in offline-to-online RL: evaluation mismatch and improvement mismatch. They propose a method to address these  mismatch issues via a combination of techniques such as value re-evaluation, alignment, and constrained fine-tuning. Overall, the results in the paper are convincing in that this approach can improve over offline policies produced from arbitrary methods.

The reviewers generally liked the paper and found their concerns addressed, so we are accepting this paper for now. The AC personally found the presentation of the paper and the ideas a bit confusing and not entirely motivated: with so many variants of the approach presented, and with some of these being a bit less concretely defined: e.g., the alignment loss for TD3, SAC are a bit ad-hoc mostly based on approximate intuition, the constrained fine-tuning loss is largely disconnected from the mismatch intuitions, the method is complex and needs dataset-specific tuning (as compared to domain-specific tuning). We encourage the authors to improve the clarity of the paper, make it clear upfront what the components of the method would be and strive to provide more guidance to researchers building on this approach.